# SMOOTHNESS AND STABILITY IN GANS

**Casey Chu**
Stanford University
caseychu@stanford.edu

**Kentaro Minami**
Preferred Networks, Inc.
minami@preferred.jp

**Kenji Fukumizu**
The Institute of Statistical Mathematics / Preferred Networks, Inc.
fukumizu@ism.ac.jp

## ABSTRACT

Generative adversarial networks, or GANs, commonly display unstable behavior during training. In this work, we develop a principled theoretical framework for understanding the stability of various types of GANs. In particular, we derive conditions that guarantee eventual stationarity of the generator when it is trained with gradient descent, conditions that must be satisfied by the divergence that is minimized by the GAN and the generator's architecture. We find that existing GAN variants satisfy some, but not all, of these conditions. Using tools from convex analysis, optimal transport, and reproducing kernels, we construct a GAN that fulfills these conditions simultaneously. In the process, we explain and clarify the need for various existing GAN stabilization techniques, including Lipschitz constraints, gradient penalties, and smooth activation functions.

## 1 INTRODUCTION: TAMING INSTABILITY WITH SMOOTHNESS

Generative adversarial networks (Goodfellow et al., 2014), or GANs, are a powerful class of generative models defined through minimax game. GANs and their variants have shown impressive performance in synthesizing various types of datasets, especially natural images. Despite these successes, the training of GANs remains quite unstable in nature, and this instability remains difficult to understand theoretically.

Since the introduction of GANs, there have been many techniques proposed to stabilize GANs training, including studies of new generator/discriminator architectures, loss functions, and regularization techniques. Notably, Arjovsky et al. (2017) proposed Wasserstein GAN (WGAN), which in principle avoids instability caused by mismatched generator and data distribution supports. In practice, this is enforced by Lipschitz constraints, which in turn motivated developments like gradient penalties (Gulrajani et al., 2017) and spectral normalization (Miyato et al., 2018). Indeed, these stabilization techniques have proven essential to achieving the latest state-of-the-art results (Karras et al., 2018; Brock et al., 2019).

On the other hand, a solid theoretical understanding of training stability has not been established. Several empirical observations point to an incomplete understanding. For example, why does applying a gradient penalty together spectral norm seem to improve performance (Miyato et al., 2018), even though in principle they serve the same purpose? Why does applying only spectral normalization with the Wasserstein loss fail (Miyato, 2018), even though the analysis of Arjovsky et al. (2017) suggests it should be sufficient? Why is applying gradient penalties effective, even outside their original context of the Wasserstein GAN (Fedus et al., 2018)?

In this work, we develop a framework to analyze the stability of GAN training that resolves these apparent contradictions and clarifies the roles of these regularization techniques. Our approach considers the *smoothness* of the loss function used. In optimization, smoothness is a well-known condition that ensures that gradient descent and its variants become stable (see e.g., Bertsekas (1999)). For example, the following well-known proposition is the starting point of our stability analysis:

**Proposition 1** (Bertsekas (1999), Proposition 1.2.3). *Suppose $f : \mathbb{R}^p \to \mathbb{R}$ is L-smooth and bounded below. Let $x_{k+1} := x_k - \frac{1}{L}\nabla f(x_k)$. Then $||\nabla f(x_k)|| \to 0$ as $k \to \infty$.*

This proposition says that under a smoothness condition on the function, gradient descent with a constant step size $\frac{1}{L}$ approaches stationarity (i.e., the gradient norm approaches zero). This is a rather weak notion of convergence, as it does not guarantee that the iterates converge to a point, and even if the iterates do converge, the limit is a stationary point and not necessarily an minimizer.

Nevertheless, empirically, not even this stationarity is satisfied by GANs, which are known to frequently destabilize and diverge during training. To diagnose this instability, we consider the smoothness of the GAN's loss function. GANs are typically framed as minimax problems of the form

$$\inf_\theta \sup_\varphi \mathcal{J}(\mu_\theta, \varphi), \tag{1}$$

where $\mathcal{J}$ is a loss function that takes a generator distribution $\mu_\theta$ and discriminator $\varphi$, and $\theta \in \mathbb{R}^p$ denotes the parameters of the generator. Unfortunately, the minimax nature of this problem makes stability and convergence difficult to analyze. To make the analysis more tractable, we define $J(\mu) = \sup_\varphi \mathcal{J}(\mu, \varphi)$, so that (1) becomes simply

$$\inf_\theta J(\mu_\theta). \tag{2}$$

This choice corresponds to the common assumption that the discriminator is allowed to reach optimality at every training step. Now, the GAN algorithm can be regarded as simply gradient descent on the $\mathbb{R}^p \to \mathbb{R}$ function $\theta \mapsto J(\mu_\theta)$, which may be analyzed using Proposition 1. In particular, if this function $\theta \mapsto J(\mu_\theta)$ satisfies the smoothness assumption, then the GAN training should be stable in that it should approach stationarity under the assumption of an optimal discriminator.

In the remainder of this paper, we investigate whether the smoothness assumption is satisfied for various GAN losses. Our analysis answers two questions:

Q1. Which existing GAN losses, if any, satisfy the smoothness condition in Proposition 1?

Q2. Are there choices of loss, regularization, or architecture that enforce smoothness in GANs?

As results of our analysis, our contributions are as follows:

1. We derive sufficient conditions for the GAN algorithm to be stationary under certain assumptions (Theorem 1). Our conditions relate to the smoothness of GAN loss used as well as the parameterization of the generator.

2. We show that most common GAN losses do not satisfy the all of the smoothness conditions, thereby corroborating their empirical instability.

3. We develop regularization techniques that enforce the smoothness conditions. These regularizers recover common GAN stabilization techniques such as gradient penalties and spectral normalization, thereby placing their use on a firmer theoretical foundation.

4. Our analysis provides several practical insights, suggesting for example the use of smooth activation functions, simultaneous spectral normalization and gradient penalties, and a particular learning rate for the generator.

## 1.1 RELATED WORK

**Divergence minimization**  Our analysis regards the GAN algorithm as minimizing a divergence between the current generator distribution and the desired data distribution, under the assumption of an optimal discriminator at every training step. This perspective originates from the earliest GAN paper, in which Goodfellow et al. (2014) show that the original minimax GAN implicitly minimizes the Jensen–Shannon divergence. Since then, the community has introduced a large number of GAN or GAN-like variants that learn generative models by implicitly minimizing various divergences, including $f$-divergences (Nowozin et al., 2016), Wasserstein distance (Arjovsky et al., 2017), and maximum-mean discrepancy (Li et al., 2015; Unterthiner et al., 2018). Meanwhile, the non-saturating GAN (Goodfellow et al., 2014) has been shown to minimize a certain Kullback–Leibler divergence (Arjovsky & Bottou, 2017). Several more theoretical works consider the topological, geometric, and convexity properties of divergence minimization (Arjovsky & Bottou, 2017; Liu et al., 2017; Bottou et al., 2018; Farnia & Tse, 2018; Chu et al., 2019), perspectives that we draw heavily upon. Sanjabi et al. (2018) also prove smoothness of GAN losses in the specific case of the regularized optimal transport loss. Their assumption for smoothness is entangled in that it involves a composite condition on generators and discriminators, while our analysis addresses them separately.

Table 1: Common GANs, their corresponding loss functions, and their optimal discriminators.

|  | Loss function $J(\mu)$ | Optimal discriminator $\Phi_\mu(x)$ |
|---|---|---|
| Minimax GAN | $D_{\text{JS}}(\mu \,\|\, \mu_0)$ | $\frac{1}{2} \log \frac{\mu(x)}{\mu(x) + \mu_0(x)}$ |
| Non-saturating GAN | $D_{\text{KL}}(\frac{1}{2}\mu + \frac{1}{2}\mu_0 \,\|\, \mu_0)$ | $-\frac{1}{2} \log \frac{\mu_0(x)}{\mu(x) + \mu_0(x)}$ |
| Wasserstein GAN | $W_1(\mu, \mu_0)$ | $\arg\max_{f \in \text{Lip}_1} \mathbb{E}_{y \sim \mu}[f(y)] - \mathbb{E}_{y \sim \mu_0}[f(y)]$ |
| GMMN, Coulomb GAN | $\frac{1}{2}\text{MMD}^2(\mu, \mu_0)$ | $\mathbb{E}_{y \sim \mu}[K(x,y)] - \mathbb{E}_{y \sim \mu_0}[K(x,y)]$ |
| IPM-GAN | $\text{IPM}_{\mathcal{F}}(\mu, \mu_0)$ | $\arg\max_{f \in \mathcal{F}} \mathbb{E}_{y \sim \mu}[f(y)] - \mathbb{E}_{y \sim \mu_0}[f(y)]$ |

**Other approaches** Even though many analyses, including ours, operate under the assumption of an optimal discriminator, this assumption is unrealistic in practice. Li et al. (2017b) contrast this optimal discriminator dynamics with *first-order dynamics*, which assumes that the generator and discriminator use alternating gradient updates and is what is used computationally. As this is a differing approach from ours, we only briefly mention some results in this area, which typically rely on game-theoretic notions (Kodali et al., 2017; Grnarova et al., 2018; Oliehoek et al., 2018) or local analysis (Nagarajan & Kolter, 2017; Mescheder et al., 2018). Some of these results rely on continuous dynamics approximations of gradient updates; in contrast, our work focuses on discrete dynamics.

## 1.2 NOTATION

Let $\bar{\mathbb{R}} := \mathbb{R} \cup \{\infty, -\infty\}$. We let $\mathcal{P}(X)$ denote the set of all probability measures on a compact set $X \subseteq \mathbb{R}^d$. We let $\mathcal{M}(X)$ and $\mathcal{C}(X)$ denote the dual pair consisting of the set of all finite signed measures on $X$ and the set of all continuous functions $X \to \mathbb{R}$. For any statement $A$, we let $\chi\{A\}$ be $0$ if $A$ is true and $\infty$ if $A$ is false. For a Euclidean vector $x$, its Euclidean norm is denoted by $\|x\|_2$, and the operator norm of a matrix $A$ is denoted by $\|A\|_2$, i.e., $\|A\|_2 = \sup_{\|x\|_2 \leq 1} \|Ax\|_2 / \|x\|_2$. A function $f : X \to Y$ between two metric spaces is $\alpha$-Lipschitz if $d_Y(f(x_1), f(x_2)) \leq \alpha d_X(x_1, x_2)$. A function $f : \mathbb{R}^d \to \mathbb{R}$ is $\beta$-smooth if its gradients are $\beta$-Lipschitz, that is, for all $x, y \in \mathbb{R}^d$, $\|\nabla f(x) - \nabla f(y)\|_2 \leq \beta \|x - y\|_2$.

## 2 SMOOTHNESS OF GAN LOSSES

This section presents Theorem 1, which provides concise criteria for the smoothness of GAN losses.

In order to keep our analysis agnostic to the particular GAN used, let $J : \mathcal{P}(X) \to \bar{\mathbb{R}}$ be an arbitrary convex loss function, which takes a distribution over $X \subset \mathbb{R}^d$ and outputs a real number. Note that the typical minimax formulation of GANs can be recovered from just the loss function $J$ using convex duality. In particular, recall that the **convex conjugate** $J^\star : \mathcal{C}(X) \to \bar{\mathbb{R}}$ of $J$ satisfies the following remarkable duality, known as the Fenchel–Moreau theorem:

$$J^\star(\varphi) := \sup_{\mu \in \mathcal{M}(X)} \int \varphi(x)\, d\mu - J(\mu), \qquad J(\mu) = \sup_{\varphi \in \mathcal{C}(X)} \int \varphi(x)\, d\mu - J^\star(\varphi). \qquad (3)$$

Based on this duality, minimizing $J$ can be framed as the minimax problem

$$\inf_{\mu \in \mathcal{P}(X)} J(\mu) = \inf_{\mu \in \mathcal{P}(X)} \sup_{\varphi \in \mathcal{C}(X)} \int \varphi(x)\, d\mu - J^\star(\varphi) := \inf_{\mu \in \mathcal{P}(X)} \sup_{\varphi \in \mathcal{C}(X)} \mathcal{J}(\mu, \varphi), \qquad (4)$$

recovering the well-known adversarial formulation of GANs. We now define the notion of an optimal discriminator for an arbitrary loss function $J$, based on this convex duality:

**Definition 1** (Optimal discriminator). *Let $J : \mathcal{M}(X) \to \bar{\mathbb{R}}$ be a convex, l.s.c., proper function. An **optimal discriminator** for a probability distribution $\mu \in \mathcal{P}(X)$ is a continuous function $\Phi_\mu : X \to \mathbb{R}$ that attains the maximum of the second equation in* (3)*, i.e., $J(\mu) = \int \Phi_\mu(x)\, d\mu - J^\star(\Phi_\mu)$.*

This definition recovers the optimal discriminators of many existing GAN and GAN-like algorithms (Farnia & Tse, 2018; Chu et al., 2019), most notably those in Table 1. Our analysis will apply to any algorithm in this family of algorithms. See Appendix B for more details on this perspective.

We also formalize the notion of a family of generators:

**Definition 2** (Family of generators). *A **family of generators** is a set of pushforward probability measures $\{\mu_\theta = f_{\theta\#}\omega : \theta \in \mathbb{R}^p\}$, where $\omega$ is a fixed probability distribution on $Z$ (the **latent variable**) and $f_\theta : Z \to X$ is a measurable function (the **generator**).*

Now, in light of Proposition 1, we are interested in the smoothness of the mapping $\theta \mapsto J(\mu_\theta)$, which would guarantee the stationarity of gradient descent on this objective, which in turn implies stationarity of the GAN algorithm under the assumption of an optimal discriminator. The following theorem is our central result, which decomposes the smoothness of $\theta \mapsto J(\mu_\theta)$ into conditions on optimal discriminators and the family of generators.

**Theorem 1** (Smoothness decomposition for GANs). *Let $J : \mathcal{M}(X) \to \bar{\mathbb{R}}$ be a convex function whose optimal discriminators $\Phi_\mu : X \to \mathbb{R}$ satisfy the following regularity conditions:*

**(D1)** $x \mapsto \Phi_\mu(x)$ *is $\alpha$-Lipschitz,*

**(D2)** $x \mapsto \nabla_x \Phi_\mu(x)$ *is $\beta_1$-Lipschitz,*

**(D3)** $\mu \mapsto \nabla_x \Phi_\mu(x)$ *is $\beta_2$-Lipschitz w.r.t. the 1-Wasserstein distance.*

*Also, let $\mu_\theta = f_{\theta\#}\omega$ be a family of generators that satisfies:*

**(G1)** $\theta \mapsto f_\theta(z)$ *is $A$-Lipschitz in expectation for $z \sim \omega$, i.e., $\mathbb{E}_{z\sim\omega}[\|f_{\theta_1}(z) - f_{\theta_2}(z)\|_2] \leq A\|\theta_1 - \theta_2\|_2$, and*

**(G2)** $\theta \mapsto D_\theta f_\theta(z)$ *is $B$-Lipschitz in expectation for $z \sim \omega$, i.e., $\mathbb{E}_{z\sim\omega}[\|D_{\theta_1}f_{\theta_1}(z) - D_{\theta_2}f_{\theta_2}(z)\|_2] \leq B\|\theta_1 - \theta_2\|_2$.*

*Then $\theta \mapsto J(\mu_\theta)$ is $L$-smooth, with $L = \alpha B + A^2(\beta_1 + \beta_2)$.*

Theorem 1 connects the smoothness properties of the loss function $J$ with the smoothness properties of the optimal discriminator $\Phi_\mu$, and once paired with Proposition 1, it suggests a quantitative value $\frac{1}{L}$ for a stable generator learning rate. In order to obtain claims of stability for practically sized learning rates, it is important to tightly bound the relevant constants.

In Sections 4 to 6, we carefully analyze which GAN losses satisfy (D1), (D2), and (D3), and with what constants. We summarize our results in Table 2: it turns out that none of the listed losses, except for one, satisfy (D1), (D2), and (D3) simultaneously with a finite constant. The MMD-based loss satisfies the three conditions, but its constant for (D1) grows as $\alpha = O(\sqrt{d})$, which is an unfavorable dependence on the data dimension $d$ that forces an unacceptably small learning rate. See for complete details of each condition. This failure of existing GANs to satisfy the stationarity conditions corroborates the observed instability of GANs.

Table 2: Regularity of common GAN losses.

|  | (D1) | (D2) | (D3) |
|---|---|---|---|
| Minimax GAN | ✗ | ✗ | ✗ |
| Non-saturating GAN | ✗ | ✗ | ✗ |
| Wasserstein GAN | ✓ | ✗ | ? |
| IPM$_{\mathcal{S}}$ | ✗ | ✓ | ? |
| MMD$^2$ | ✓* | ✓ | ✓ |

Theorem 1 decomposes smoothness into conditions on the generator and conditions on the discriminator, allowing a clean separation of concerns. In this paper, we focus on the discriminator conditions (D1), (D2), and (D3) and only provide an extremely simple example of a generator that satisfies (G1) and (G2), in Section 7. Because analysis of the generator conditions may become quite complicated and will vary with the choice of architecture considered (feedforward, convolutional, ResNet, etc.), we leave a detailed analysis of the generator conditions (G1) and (G2) as a promising avenue for future work. Indeed, such analyses may lead to new generator architectures or generator regularization techniques that stabilize GAN training.

## 3 ENFORCING SMOOTHNESS WITH INF-CONVOLUTIONS

In this section, we present a generic regularization technique that imposes the three conditions sufficient for stable learning on an arbitrary loss function $J$, thereby stabilizing training. In Section 2, we observe that the Wasserstein, IPM, and MMD losses respectively satisfy (D1), (D2), and (D3) individually, but not all of of them at the same time. Using techniques from convex analysis, we convert these three GAN losses into three regularizers that, when applied simultaneously, causes the resulting loss to satisfy all the three conditions. Here, we only outline the technique; the specifics of each case are deferred to Sections 4 to 6.

We start with an arbitrary base loss function $J$ to be regularized. Next, we take an existing GAN loss that satisfies the desired regularity condition and convert it into a **regularizer** function $R :\mathcal{M}(X) \to \bar{\mathbb{R}}$. Then, we consider $J \oplus R$, which denotes the **inf-convolution** defined as

$$(J \oplus R)(\xi) = \inf_{\tilde{\xi} \in \mathcal{M}(X)} J(\tilde{\xi}) + R(\xi - \tilde{\xi}). \tag{5}$$

This new function $J \oplus R$ inherits the regularity of $R$, making it a stable candidate as a GAN loss. Moreover, because the inf-convolution is a commutative operation, we can sequentially apply multiple regularizers $R_1$, $R_2$, and $R_3$ without destroying the added regularity. In particular, if we carefully choose functions $R_1$, $R_2$, and $R_3$, then $\tilde{J} = J \oplus R_1 \oplus R_2 \oplus R_3$ will satisfy (D1), (D2), and (D3) simultaneously. Moreover, under some technical assumptions, this composite function $\tilde{J}$ inherits the original minimizers of $J$, making it a sensible GAN loss:

**Proposition 2** (Invariance of minimizers). *Let $R_1(\xi) := \|\xi\|_{\mathrm{KR}}$, $R_2(\xi) := \|\xi\|_{\mathcal{S}*}$, and $R_3(\xi) := \frac{1}{2}\|\hat{\xi}\|_{\mathcal{H}}^2$ be the three regularizers defined by (8), (12), and (19) respectively. Assume that $J :\mathcal{M}(X) \to \bar{\mathbb{R}}$ has a unique minimizer at $\mu_0$ with $J(\mu_0) = 0$, and $J(\mu) \geq c\|\hat{\mu} - \hat{\mu}_0\|_{\mathcal{H}}$ for some $c > 0$. Then the inf-convolution $\tilde{J} = J \oplus R_1 \oplus R_2 \oplus R_3$ has a unique minimizer at $\mu_0$ with $\tilde{J}(\mu_0) = 0$.*

The duality formulation (4) provides a practical method for minimizing this composite function. We leverage the duality relation $(J \oplus R_1 \oplus R_2 \oplus R_3)^\star = J^\star + R_1^\star + R_2^\star + R_3^\star$ and apply (4):

$$\inf_{\mu} (J \oplus R_1 \oplus R_2 \oplus R_3)(\mu) = \inf_{\mu} \sup_{\varphi} \int \varphi \, d\mu - J^\star(\varphi) - R_1^\star(\varphi) - R_2^\star(\varphi) - R_3^\star(\varphi) \tag{6}$$

$$= \inf_{\mu} \sup_{\varphi} \mathcal{J}(\mu, \varphi) - R_1^\star(\varphi) - R_2^\star(\varphi) - R_3^\star(\varphi). \tag{7}$$

This minimax problem can be seen as a GAN whose discriminator objective has three added regularization terms.

The concrete form of these regularizers are summarized in Table 3. Notably, we observe that we recover standard techniques for stabilizing GANs:

- (D1) is enforced by **Lipschitz constraints** (i.e., **spectral normalization**) on the discriminator.
- (D2) is enforced by spectral normalization and a choice of Lipschitz, smooth activation functions for the discriminator.

Table 3: Smoothness-inducing regularizers and their convex conjugates. To enforce a regularity condition on a loss function $J$, we take a source loss that satisfies it and view it as a regularizer $R$. We then consider the inf-convolution $J \oplus R$, which corresponds to an added regularization term $R^\star$ on the discriminator $\varphi$. These regularization terms correspond to existing GAN techniques.

| Purpose | Source loss | $R(\xi)$ | $R^\star(\varphi)$ | GAN techniques |
|---------|-------------|----------|---------------------|----------------|
| (D1) | $W_1$ | $\|\xi\|_{\mathrm{KR}}$ | $\chi\{\|\varphi\|_{\mathrm{Lip}} \leq 1\}$ | spectral norm |
| (D2) | $\mathrm{IPM}_{\mathcal{S}}$ | $\|\xi\|_{\mathcal{S}*}$ | $\chi\{\varphi \in \mathcal{S}\}$ | smooth activations, spectral norm |
| (D3) | $\mathrm{MMD}^2$ | $\frac{1}{2}\|\xi\|_{\mathcal{H}}^2$ | $\frac{1}{2}\|\varphi\|_{\mathcal{H}}^2$ | gradient penalties |

- (D3) is enforced by **gradient penalties** on the discriminator.

Our analysis therefore puts these regularization techniques on a firm theoretical foundation (Proposition 1 and Theorem 1) and provides insight into their function.

## 4 Enforcing (D1) with Lipschitz constraints

In this section, we show that enforcing (D1) leads to techniques and notions commonly used to stabilize GANs, including the Wasserstein distance, Lipschitz constraints and spectral normalization. Recall that (D1) demands that the optimal discriminator $\Phi_\mu$ is Lipschitz:

**(D1)** $x \mapsto \Phi_\mu(x)$ *is $\alpha$-Lipschitz for all $\mu \in \mathcal{P}(X)$, i.e., $|\Phi_\mu(x) - \Phi_\mu(y)| \leq \alpha ||x - y||_2$.*

If $\Phi_\mu$ is differentiable, this is equivalent to that the optimal discriminator has a gradient with bounded norm. This is a sensible criterion, since a discriminator whose gradient norm is too large may push the generator too hard and destabilize its training.

To check (D1), the following proposition shows that it suffices to check whether $|J(\mu) - J(\nu)| \leq \alpha W_1(\mu, \nu)$ for all distributions $\mu, \nu$:

**Proposition 3.** *(D1) holds if and only if $J$ is $\alpha$-Lipschitz w.r.t. the Wasserstein-1 distance.*

Arjovsky et al. (2017) show that this property does not hold for common divergences based on the Kullback–Leibler or Jensen–Shannon divergence, while it does hold for the Wasserstein-1 distance. Indeed, it is this desirable property that motivates their introduction of the Wasserstein GAN. Framed in our context, their result is summarized as follows:

**Proposition 4.** *The minimax and non-saturating GAN losses do not satisfy (D1) for some $\mu_0$.*

**Proposition 5.** *The Wasserstein GAN loss satisfies (D1) with $\alpha = 1$ for any $\mu_0$.*

Our stability analysis therefore deepens the analysis of Arjovsky et al. (2017) and provides an alternative reason that the Wasserstein distance is desirable as a metric: it is part of a sufficient condition that ensures stationarity of gradient descent.

### 4.1 From Wasserstein distance to Lipschitz constraints

Having identified the Wasserstein GAN loss as one that satisfies (D1), we next follow the strategy outlined in Section 3 to convert it into a regularizer for an arbitrary loss function. Towards this, we define the regularizer $R_1 : \mathcal{M}(X) \to \bar{\mathbb{R}}$ and compute its convex conjugate $R_1^\star : \mathcal{C}(X) \to \bar{\mathbb{R}}$:

$$R_1(\xi) := \alpha \|\xi\|_{\mathrm{KR}} = \alpha \sup_{\substack{f \in \mathcal{C}(X) \\ \|f\|_{\mathrm{Lip}} \leq 1}} \int f \, d\xi, \qquad R_1^\star(\varphi) = \begin{cases} 0 & \|\varphi\|_{\mathrm{Lip}} \leq \alpha \\ \infty & \text{otherwise.} \end{cases} \tag{8}$$

This norm is the **Kantorovich–Rubinstein norm** (KR norm), which extends the Wasserstein-1 distance to $\mathcal{M}(X)$; it holds that $\|\mu - \nu\|_{\mathrm{KR}} = W_1(\mu, \nu)$ for $\mu, \nu \in \mathcal{P}(X)$. Then, its inf-convolution with an arbitrary function inherits the Lipschitz property held by $R_1$:

**Proposition 6** (Pasch–Hausdorff). *Let $J : \mathcal{M}(X) \to \bar{\mathbb{R}}$ be a function, and define $\tilde{J} := J \oplus R_1$. Then $\tilde{J}$ is $\alpha$-Lipschitz w.r.t. the distance induced by the KR norm, and hence the Wasserstein-1 distance when restricted to $\mathcal{P}(X)$.*

Due to Proposition 3, we now obtain a transformed loss function $\tilde{J}$ that automatically satisfies (D1). This function is a generalization of the **Pasch–Hausdorff envelope** (see Chapter 9 in Rockafeller & Wets (1998)), also known as Lipschitz regularization or the McShane–Whitney extension (McShane, 1934; Whitney, 1934; Kirszbraun, 1934; Hiriart-Urruty, 1980).

The convex conjugate computation in (8) shows that $\tilde{J}$ can be minimized in practice by imposing Lipschitz constraints on discriminators. Indeed, by (4),

$$\inf_\mu \left( J \oplus \alpha \|\cdot\|_{\mathrm{KR}} \right)(\mu) = \inf_\mu \sup_\varphi \mathbb{E}_{x \sim \mu}[\varphi(x)] - J^\star(\varphi) - \chi\{\|\varphi\|_{\mathrm{Lip}} \leq \alpha\} \tag{9}$$

$$= \inf_{\mu} \sup_{\varphi: \|\varphi\|_{\mathrm{Lip}} \leq \alpha} \mathcal{J}(\mu, \varphi). \tag{10}$$

Farnia & Tse (2018) consider this loss $\tilde{J}$ in the special case of an $f$-GAN with $J(\mu) = D_f(\mu, \mu_0)$; they showed that minimizing $\tilde{J}$ corresponds to training a $f$-GAN normally but constraining the discriminator to be $\alpha$-Lipschitz. We show that this technique is in fact generic for any $J$: minimizing the transformed loss can be achieved by training the GAN as normal, but imposing a Lipschitz constraint on the discriminator.

Our analysis therefore justifies the use of Lipschitz constraints, such as spectral normalization (Miyato et al., 2018) and weight clipping (Arjovsky & Bottou, 2017), for general GAN losses. However, Theorem 1 also suggests that applying only Lipschitz constraints may not be enough to stabilize GANs, as (D1) alone does not ensure that the GAN objective is smooth.

## 5 ENFORCING (D2) WITH DISCRIMINATOR SMOOTHNESS

(D2) demands that the optimal discriminator $\Phi_\mu$ is smooth:

**(D2)** $x \mapsto \nabla \Phi_\mu(x)$ is $\beta_1$-Lipschitz for all $\mu \in \mathcal{P}(X)$, i.e., $\|\nabla \Phi_\mu(x) - \nabla \Phi_\mu(y)\|_2 \leq \beta_1 \|x - y\|_2$.

Intuitively, this says that for a fixed generator $\mu$, the optimal discriminator $\Phi_\mu$ should not provide gradients that change too much spatially.

Although the Wasserstein GAN loss (D1), we see that it, along with the minimax GAN and the non-saturating GAN, do not satisfy (D2):

**Proposition 7.** *The Wasserstein, minimax, and non-saturating GAN losses do not satisfy (D2) for some $\mu_0$.*

We now construct a loss that by definition satisfies (D2). Let $\mathcal{S}$ be the class of 1-smooth functions, that is, for which $\|\nabla f(x) - \nabla f(y)\|_2 \leq \|x - y\|_2$, and consider the **integral probability metric** (IPM) (Müller, 1997) w.r.t. $\mathcal{S}$, defined by

$$\mathrm{IPM}_{\mathcal{S}}(\mu, \nu) := \sup_{f \in \mathcal{S}} \int f \, d\mu - \int f \, d\nu. \tag{11}$$

The optimal discriminator for the loss $\mathrm{IPM}_{\mathcal{S}}(\mu, \mu_0)$ is the function that maximizes the supremum in the definition. This function by definition belongs to $\mathcal{S}$ and therefore is 1-smooth. Hence, this IPM loss satisfies (D2) with $\beta_1 = 1$ by construction.

### 5.1 FROM INTEGRAL PROBABILITY METRIC TO SMOOTH DISCRIMINATORS

Having identified the IPM-based loss as one that satisfies (D2), we next follow the strategy outlined in Section 3 to convert it into a regularizer for an arbitrary loss function. To do so, we define a regularizer $R_2 : \mathcal{M}(X) \to \bar{\mathbb{R}}$ and compute its convex conjugate $R_2^\star : \mathcal{C}(X) \to \bar{\mathbb{R}}$:

$$R_2(\xi) := \beta_1 \|\xi\|_{\mathcal{S}^*} = \beta_1 \sup_{f \in \mathcal{S}} \int f \, d\xi, \qquad R_2^\star(\varphi) = \begin{cases} 0 & \varphi \in \beta_1 \mathcal{S} \\ \infty & \text{otherwise.} \end{cases} \tag{12}$$

The norm is the **dual norm** to $\mathcal{S}$, which extends the IPM to signed measures; it holds that $\mathrm{IPM}_{\mathcal{S}}(\mu, \nu) = \|\mu - \nu\|_{\mathcal{S}^*}$ for $\mu, \nu \in \mathcal{P}(X)$. Similar to the situation in the previous section, inf-convolution preserves the smoothness property of $R_2$:

**Proposition 8.** *Let $J : \mathcal{M}(X) \to \bar{\mathbb{R}}$ be a convex, proper, lower semicontinuous function, and define $\tilde{J} := J \oplus R_2$. Then the optimal discriminator for $\tilde{J}$ is $\beta_1$-smooth.*

Applying (4) and (12), we see that we can minimize this transformed loss function by restricting the family of discriminators to only $\beta_1$-smooth discriminators:

$$\inf_{\mu} (J \oplus \beta_1 \|\cdot\|_{\mathcal{S}^*})(\mu) = \inf_{\mu} \sup_{\varphi} \mathbb{E}_{x \sim \mu}[\varphi(x)] - J^\star(\varphi) - \chi\{\varphi \in \beta_1 \mathcal{S}\} \tag{13}$$

$$= \inf_{\mu} \sup_{\varphi \in \beta_1 \mathcal{S}} \mathcal{J}(\mu, \varphi). \tag{14}$$

In practice, we can enforce this by applying spectral normalization (Miyato et al., 2018) and using a Lipschitz, smooth activation function such as ELU (Clevert et al., 2016) or sigmoid.

**Proposition 9.** *Let $f : \mathbb{R}^d \to \mathbb{R}$ be a neural network consisting of $k$ layers whose linear transformations have spectral norm $1$ and whose activation functions are $1$-Lipschitz and $1$-smooth. Then $f$ is $k$-smooth.*

## 6 ENFORCING (D3) WITH GRADIENT PENALTIES

(D3) is the following smoothness condition:

**(D3)** $\mu \mapsto \nabla \Phi_\mu(x)$ *is* $\beta_2$-*Lipschitz for any* $x \in X$, *i.e.,* $\|\nabla \Phi_\mu(x) - \nabla \Phi_\nu(x)\|_2 \leq \beta_2 W_1(\mu, \nu)$.

(D3) requires that the gradients of the optimal discriminator do not change too rapidly in response to changes in $\mu$. Indeed, if the discriminator's gradients are too sensitive to changes in the generator, the generator may not be able to accurately follow those gradients as it updates itself using a finite step size. In finite-dimensional optimization of a function $f : \mathbb{R}^d \to \mathbb{R}$, this condition is analogous to $f$ having a Lipschitz gradient.

We now present an equivalent characterization of (D3) that is easier to check in practice. We define the **Bregman divergence** of a convex function $J : \mathcal{M}(X) \to \bar{\mathbb{R}}$ by

$$\mathfrak{D}_J(\nu, \mu) := J(\nu) - J(\mu) - \int \Phi_\mu(x) \, d(\nu - \mu), \tag{15}$$

where $\Phi_\mu$ is the optimal discriminator for $J$ at $\mu$. Then, (D3) is characterized in terms of the Bregman divergence and the KR norm as follows:

**Proposition 10.** *Let $J : \mathcal{M}(X) \to \mathbb{R}$ be a convex function. Then $J$ satisfies (D3) if and only if $\mathfrak{D}_J(\nu, \mu) \leq \frac{\beta_2}{2} \|\mu - \nu\|_{\mathrm{KR}}^2$ for all $\mu, \nu \in \mathcal{M}(X)$.*

It is straightforward to compute the Bregman divergence corresponding to several popular GANs:

$$\mathfrak{D}_{D_{\mathrm{JS}}(\cdot \| \mu_0)}(\nu, \mu) = D_{\mathrm{KL}}(\tfrac{1}{2}\nu + \tfrac{1}{2}\mu_0 \| \tfrac{1}{2}\mu + \tfrac{1}{2}\mu_0) + \tfrac{1}{2}D_{\mathrm{KL}}(\nu \| \mu), \tag{16}$$

$$\mathfrak{D}_{D_{\mathrm{KL}}(\frac{1}{2}\cdot + \frac{1}{2}\mu_0 \| \mu_0)}(\nu, \mu) = D_{\mathrm{KL}}(\tfrac{1}{2}\nu + \tfrac{1}{2}\mu_0 \| \tfrac{1}{2}\mu + \tfrac{1}{2}\mu_0), \tag{17}$$

$$\mathfrak{D}_{\frac{1}{2}\mathrm{MMD}^2(\cdot, \mu_0)}(\nu, \mu) = \tfrac{1}{2}\mathrm{MMD}^2(\nu, \mu). \tag{18}$$

The first two Bregman divergences are not bounded above by $\|\mu - \nu\|_{\mathrm{KR}}^2$ for reasons similar to those discussed in Section 4, and hence:

**Proposition 11.** *The minimax and non-saturating GAN losses do not satisfy (D3) for some $\mu_0$.*

Even so, the Bregman divergence for the non-saturating loss is always less than that of the minimax GAN, suggesting that the non-saturating loss should be stable in more situations than the minimax GAN. On the other hand, the MMD-based loss (Li et al., 2015) does satisfy (D3) when its kernel is the Gaussian kernel $K(x, y) = e^{-\pi \|x - y\|^2}$:

**Proposition 12.** *The MMD loss with Gaussian kernel satisfies (D3) with $\beta_2 = 2\pi$ for all $\mu_0$.*

### 6.1 FROM MAXIMUM MEAN DISCREPANCY TO GRADIENT PENALTIES

Having identified the MMD-based loss as one that satisfies (D3), we next follow the strategy outlined in Section 3 to convert it into a regularizer for an arbitrary loss function. To do so, we define the regularizer $R_3 : \mathcal{M}(X) \to \bar{\mathbb{R}}$ and compute its convex conjugate $R_3^\star : \mathcal{C}(X) \to \bar{\mathbb{R}}$:

$$R_3(\xi) := \frac{\beta_2}{4\pi} \|\hat{\xi}\|_{\mathcal{H}}^2, \qquad R_3^\star(\varphi) = \frac{\pi}{\beta_2} \|\varphi\|_{\mathcal{H}}^2. \tag{19}$$

The norm is the norm of a **reproducing kernel Hilbert space norm** (RKHS) $\mathcal{H}$ with Gaussian kernel; this norm extends the MMD to signed measures, as it holds that $\mathrm{MMD}(\mu, \nu) = \|\hat{\mu} - \hat{\nu}\|_{\mathcal{H}}$ for $\mu, \nu \in \mathcal{P}(X)$. Here, $\hat{\xi} = \int K(x, \cdot) \, \xi(dx) \in \mathcal{H}$ denotes the *mean embedding* of a signed measure $\xi \in \mathcal{M}(X)$; we also adopt the convention that $\|\varphi\|_{\mathcal{H}} = \infty$ if $\varphi \notin \mathcal{H}$. Similar to the situation in the previous sections, inf-convolution preserves the smoothness property of $R_3$:

**Proposition 13** (Moreau–Yosida regularization). *Suppose $J : \mathcal{M}(X) \to \bar{\mathbb{R}}$ is convex, and define $\tilde{J} := J \oplus R_3$. Then $\tilde{J}$ is convex, and $\mathfrak{D}_{\tilde{J}}(\nu, \mu) \leq \frac{\beta_2}{2} \|\mu - \nu\|_{\mathrm{KR}}^2$.*

By Proposition 10, this transformed loss function satisfies (D3), having inherited the regularity properties of the squared MMD. This transformed function is a generalization of **Moreau–Yosida regularization** or the Moreau envelope (see Chapter 1 in Rockafeller & Wets (1998)). It is well-known that in the case of a function $f : \mathbb{R}^n \to \mathbb{R}$, this regularization results in a function with Lipschitz gradients, so it is unsurprising that this property carries over to the infinite-dimensional case.

Applying (4) and (19), we see that the transformed loss function can be minimized as a GAN by implementing an RKHS squared norm penalty on the discriminator:

$$\inf_{\mu} (J \oplus \tfrac{\beta_2}{4\pi} || \cdot ||_{\mathcal{H}}^2)(\mu) = \inf_{\mu} \sup_{\varphi} \mathbb{E}_{x \sim \mu}[\varphi(x)] - J^{\star}(\varphi) - \tfrac{\pi}{\beta_2} ||\varphi||_{\mathcal{H}}^2. \tag{20}$$

Computationally, the RKHS norm is difficult to evaluate. We propose taking advantage of the following infinite series representation of $||f||_{\mathcal{H}}^2$ in terms of the derivatives of $f$ (Fasshauer & Ye, 2011; Novak et al., 2018):

**Proposition 14.** *Let $\mathcal{H}$ be an RKHS with the Gaussian kernel $K(x,y) = e^{-\pi||x-y||^2}$. Then for $f \in \mathcal{H}$,*

$$||f||_{\mathcal{H}}^2 = \sum_{k=0}^{\infty} (4\pi)^{-k} \sum_{k_1 + \cdots + k_d = k} \frac{1}{\prod_{i=1}^{d} k_i!} ||\partial_{x_1}^{k_1} \cdots \partial_{x_d}^{k_d} f||_{L^2(\mathbb{R}^d)}^2 \tag{21}$$

$$= ||f||_{L^2(\mathbb{R}^d)}^2 + \tfrac{1}{4\pi} ||\nabla f||_{L^2(\mathbb{R}^d)}^2 + \tfrac{1}{16\pi^2} ||\nabla^2 f||_{L^2(\mathbb{R}^d)}^2 + \textit{other terms}. \tag{22}$$

In an ideal world, we would use this expression as a penalty on the discriminator to enforce (D3). Of course, as an infinite series, this formulation is computationally impractical. However, the first two terms are very close to common GAN techniques like gradient penalties (Gulrajani et al., 2017) and penalizing the output of the discriminator (Karras et al., 2018). We therefore interpret these common practices as partially applying the penalty given by the RKHS norm squared, approximately enforcing (D3). We view the choice of only using the leading terms as a disadvantageous but practical necessity.

Interestingly, according to our analysis, gradient penalties and spectral normalization are not interchangeable, even though both techniques were designed to constrain the Lipschitz constant of the discriminator. Instead, our analysis suggests that they serve different purposes: gradient penalties enforce the variational smoothness (D3), while spectral normalization enforces Lipschitz continuity (D1). This demystifies the puzzling observation of Miyato (2018) that GANs using only spectral normalization with a WGAN loss do not seem to train well; it also explains why using both spectral normalization and a gradient penalty is a reasonable strategy. It also motivates the use of gradient penalties applied to losses other than the Wasserstein loss (Fedus et al., 2018).

## 7 VERIFYING THE THEORETICAL LEARNING RATE

In this section, we empirically test the theoretical learning rate given by Theorem 1 and Proposition 1 as well as our regularization scheme (7) based on inf-convolutions. We approximately implement our composite regularization scheme (7) on a trivial base loss of $J(\mu) = \chi\{\mu = \mu_0\}$ by alternating stochastic gradient steps on

$$\inf_{\mu} \sup_{\varphi} \mathbb{E}_{x \sim \mu}[\varphi(x)] - \mathbb{E}_{x \sim \mu_0}[\varphi(x)] - \frac{\pi}{\beta_2} \mathbb{E}_{x \sim \tilde{\mu}}\Big[\varphi(x)^2 + \frac{1}{4\pi} ||\nabla\varphi(x)||^2\Big], \tag{23}$$

where $\tilde{\mu}$ is a random interpolate between samples from $\mu$ and $\mu_0$, as used in Gulrajani et al. (2017). The regularization term is a truncation of the series for the squared RKHS norm (22) and approximately enforces (D3). The discriminator is a 7-layer convolutional neural network with spectral normalization[1] and ELU activations, an architecture that enforces (D1) and (D2). We include a final scalar multiplication by $\alpha$ so that by Proposition 9, $\beta_1 = 7\alpha$. We take two discriminator steps for every generator step, to better approximate our assumption of an optimal discriminator.

---

[1]Whereas Miyato et al. (2018) divide each layer by the spectral norm of the convolutional kernel, we divide by the spectral norm of the convolutional operator itself, computed using the same power iteration algorithm applied to the operator. This is so that the layers are truly 1-Lipschitz, which is critical for our theory.

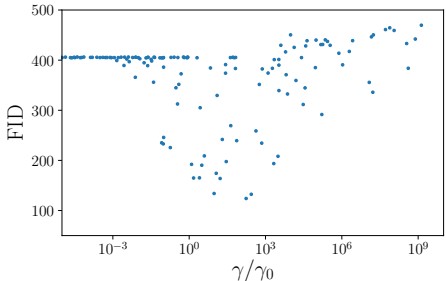

Figure 1: FID of simple particle generators with various learning rates. The horizontal axis shows the ratio $\gamma/\gamma_0$ between the actual learning rate $\gamma$ and the theoretical learning rate $\gamma_0$ suggested by Theorem 1 and Proposition 1. The vertical axis shows the FID after 100,000 SGD iterations.

For the generator, we use an extremely simple particle-based generator which satisfies (G1) and (G2), in order to minimize the number of confounding factors in our experiment. Let $\omega$ be the discrete uniform distribution on $Z = \{1, \ldots, N\}$. For an $N \times d$ matrix $\theta$ and $z \in Z$, define $f_\theta : Z \to \mathbb{R}^d$ so that $f_\theta(z)$ is the $z$th row of $\theta$. The particle generator $\mu_\theta = f_{\theta\#}\omega$ satisfies (G1) with $A = \frac{1}{\sqrt{N}}$, since

$$\mathbb{E}_z[\|f_\theta(z) - f_{\theta'}(z)\|_2] = \frac{1}{N}\sum_{z=1}^{n} \|\theta_z - \theta'_z\|_2 \leq \frac{1}{\sqrt{N}}\|\theta - \theta'\|_{\mathrm{F}}, \tag{24}$$

and it satisfies (G2) with $B = 0$, since $D_\theta f_\theta(z)$ is constant w.r.t. $\theta$. With this setup, Theorem 1 suggests a theoretical learning rate of

$$\gamma_0 = \frac{1}{L} = \frac{1}{\alpha B + A^2(\beta_1 + \beta_2)} = \frac{N}{7\alpha + \beta_2}. \tag{25}$$

We randomly generated hyperparameter settings for the Lipschitz constant $\alpha$, the smoothness constant $\beta_2$, the number of particles $N$, and the learning rate $\gamma$. We trained each model for 100,000 steps on CIFAR-10 and evaluate each model using the Fréchet Inception Distance (FID) of Heusel et al. (2017). We hypothesize that stability is correlated with image quality; Figure 1 plots the FID for each hyperparameter setting in terms of the ratio of the true learning rate $\gamma$ and the theoretically motivated learning rate $\gamma_0$. We find that the best FID scores are obtained in the region where $\gamma/\gamma_0$ is between 1 and 1000. For small learning rates $\gamma/\gamma_0 \ll 1$, we observe that the convergence is too slow to make a reasonable progress on the objective, whereas as the learning rate gets larger $\gamma/\gamma_0 \gg 1$, we observe a steady increase in FID, signalling unstable behavior. It also makes sense that learning rates slightly above the optimal rate produce good results, since our theoretical learning rate is a conservative lower bound. Note that our intention is to test our theory, not to generate good images, which is difficult due to our weak choice of generator. Overall, this experiment shows that our theory and regularization scheme are sensible.

## 8 FUTURE WORK

**Inexact gradient descent** In this paper, we employed several assumptions in order to regard the GAN algorithm as gradient descent. However, real-world GAN algorithms must be treated as "inexact" descent algorithms. As such, future work includes: (i) relaxing the optimal discriminator assumption (cf. Sanjabi et al. (2018)) or providing a stability result for discrete simultaneous gradient descent (cf. continuous time analysis in Nagarajan & Kolter (2017); Mescheder et al. (2018)), (ii) addressing stochastic approximations of gradients (i.e., SGD), and (iii) providing error bounds for the truncated gradient penalty used in (23).

**Generator architectures** Another important direction of research is to seek more powerful generator architectures that satisfy our smoothness assumptions (G1) and (G2). In practice, generators are often implemented as deep neural networks, and involve some specific architectures such as deconvolution layers (Radford et al., 2015) and residual blocks (e.g., Gulrajani et al. (2017); Miyato et al. (2018)). In this paper, we did not provide results on the smoothness of general classes of generators, since our focus is to analyze stability properties influenced by the choice of loss function $J$ (and therefore optimal discriminators). However, our conditions (G1) and (G2) shed light on how to obtain smoothly parameterized neural networks, which is left for future work.

ACKNOWLEDGMENTS

We would like to thank Kohei Hayashi, Katsuhiko Ishiguro, Masanori Koyama, Shin-ichi Maeda, Takeru Miyato, Masaki Watanabe, and Shoichiro Yamaguchi for helpful discussions.

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

## A  INF-CONVOLUTION IN $\mathbb{R}^d$

To gain intuition on the inf-convolution, we present a finite-dimensional analogue of the techniques in Section 3. For simplicity of presentation, we will omit any regularity conditions (e.g., lower semicontinuity). We refer readers to Chapter 12 of Bauschke & Combettes (2011) for a detailed introduction.

Let $J$ and $R$ be convex functions on $\mathbb{R}^d$. The inf-convolution of $J$ and $R$ is a function $J \star R$ defined as

$$(J \oplus R)(x) := \inf_{z \in \mathbb{R}^d} J(z) + R(x - z).$$

The inf-convolution is often called the epigraphic sum since the epigraph of $J \star R$ coincides with the Minkowski sum of epigraphs of $J$ and $R$, as Figure 2 illustrates. The inf-convolution is associative and commutative operation; that is, it is always true that $(J_1 \oplus J_2) \oplus J_3 = J_1 \oplus (J_2 \oplus J_3) =: J_1 \oplus J_2 \oplus J_3$ and $J_1 \oplus J_2 = J_2 \oplus J_1$.

There are two important special cases of inf-convolutions: The first one is the Pasch–Hausdorff envelope $J_\alpha$, which is the inf-convolution between $J$ and $\alpha\|\cdot\|_2$ ($\alpha > 0$). It is known that $J_\alpha$ becomes $\alpha$-Lipschitz. The second important example is the Moreau envelope $J^\beta = J \oplus \frac{1}{2\beta}\|\cdot\|_2^2$, i.e., the inf-convolution with the quadratic regularizer $\frac{1}{2\beta}\|\cdot\|_2^2$. The Moreau envelope $J^\beta$ is always differentiable, and the gradient of $J^\beta$ is $\beta$-Lipschitz (thus $J^\beta$ is $\beta$-smooth).

It is worth noting that the set of minimizers does not change after these two operations. More generally, we have the following result:

**Proposition 15.** *Let $J, R : \mathbb{R}^d \to \bar{\mathbb{R}}$ be proper and lower semicontinuous functions with $\min J > -\infty$ and $\min R = 0$. Suppose $R(0) = 0$ and $R(x) \geq \psi(\|x\|_2)$ for some increasing function $\psi : \mathbb{R}_{\geq 0} \to \mathbb{R}$. Then, $\min J \oplus R = \min J$ and $\arg\min J \oplus R = \arg\min J$.*

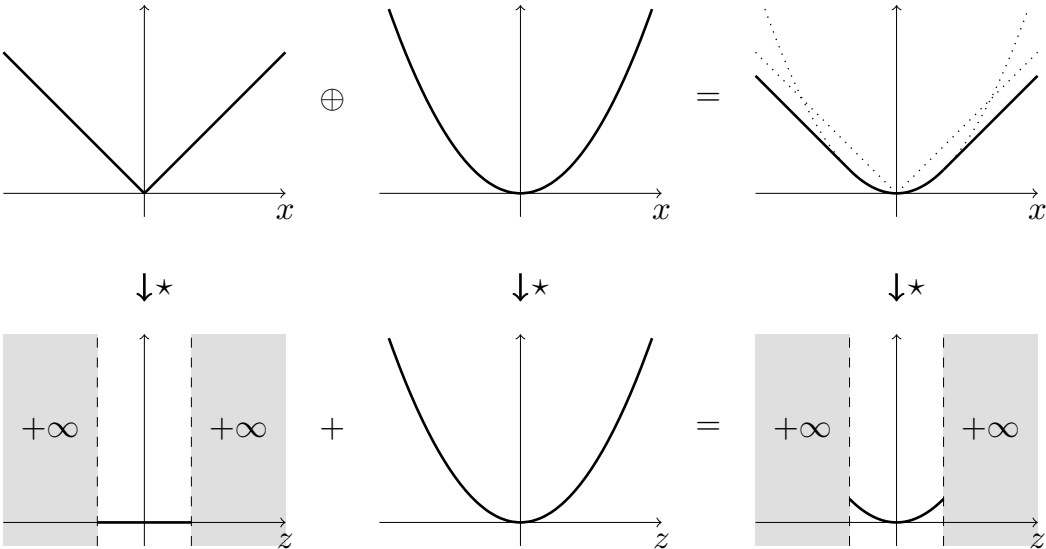

Figure 2: Illustration of inf-convolutions and their convex conjugates in $\mathbb{R}$. **Top row**: Generally, inf-convolutions inherit the regularity of their parent functions. In this example, $J_1(x) = |x|$ is 1-Lipschitz but not smooth, while $J_2(x) = x^2/2$ is 1-smooth but not Lipschitz. The inf-convolution $J_1 \oplus J_2$ is the well-known Huber function, which is both 1-Lipschitz and 1-smooth. **Bottom row**: Convex conjugates of the functions in the top row. The conjugate of $J_1 \oplus J_2$ is given as the sum of conjugates $J_1^\star(z) = \chi\{|z| \leq 1\}$ and $J_2^\star(z) = z^2/2$.

To sum up, given a function $J$, we can always construct a regularized alternative $J_\alpha^\beta$ that is $\alpha$-Lipschitz and $\beta$-smooth and has the same minimizers as $J$.

The next question is how to implement the inf-convolution in GAN-like optimization problems. For this, it is convenient to consider the convex conjugate. Recall that the Fenchel–Moreau theorem says that there is a duality between a convex function $J$ and its convex conjugate $J^\star$ as $J(x) = \sup_{z \in \mathbb{R}^d} \langle x, z \rangle - J^\star(z)$ and $J^\star(z) = \sup_{x \in \mathbb{R}^d} \langle x, z \rangle - J(x)$. The important property is that the convex conjugate of the inf-convolution is the sum of convex conjugates, that is, we always have

$$(J \oplus R)^\star(z) = J^\star(z) + R^\star(z).$$

This property can be useful for implementing the regularized objective $J_\alpha^\beta$ as follows. First, we can check that the convex conjugates of the norm and the squared norm are given as $(\|\cdot\|_2)^\star = \chi\{\|\cdot\| \leq 1\}$ and $(\frac{1}{2}\|\cdot\|_2^2)^\star = \frac{1}{2}\|\cdot\|_2^2$. Hence, we have

$$J_\alpha^\beta(x) := \left( J \oplus \alpha\|\cdot\|_2 \oplus \frac{1}{2\beta}\|\cdot\|_2^2 \right)(x) = \sup_{z: \|z\|_2 \leq \alpha} \langle x, z \rangle - J^\star(z) - \frac{\beta}{2}\|z\|_2^2,$$

which means that minimizing $J_\alpha^\beta$ can be recast in min-max problem with the norm clipping and $\ell_2$-regularization on the dual variable $z$.

## B   COMMON GAN LOSSES

For completeness and clarity, we explicitly write out the expressions for the losses listed in Table 1. For more detailed computations of optimal discriminators, see Chu et al. (2019); for more details on the convex duality interpretation, see Farnia & Tse (2018).

**Minimax GAN**   Goodfellow et al. (2014) originally proposed the minimax GAN and showed that the corresponding loss function for the minimax GAN is the Jensen–Shannon divergence, defined as

$$J(\mu) := D_{\mathrm{JS}}(\mu \,\|\, \mu_0) := \frac{1}{2}D_{\mathrm{KL}}(\mu \,\|\, \tfrac{1}{2}\mu + \tfrac{1}{2}\mu_0) + \frac{1}{2}D_{\mathrm{KL}}(\mu_0 \,\|\, \tfrac{1}{2}\mu + \tfrac{1}{2}\mu_0),$$

where $\mu_0 \in \mathcal{P}(X)$ is a fixed probability measure (usually the empirical measure of the data), and $D_{\mathrm{KL}}(\mu \,\|\, \nu)$ is the Kullback–Leibler divergence between $\mu$ and $\nu$. The optimal discriminator in the sense of Definition 1 is given as

$$\Phi_\mu(x) = \frac{1}{2} \log \frac{d\mu}{d(\mu + \mu_0)}(x),$$

where $\frac{d\mu}{d(\mu+\mu_0)}$ is the Radon–Nikodym derivative. If $\mu$ and $\mu_0$ have densities $\mu(x)$ and $\mu_0(x)$, then

$$\frac{d\mu}{d(\mu + \mu_0)}(x) = \frac{\mu(x)}{\mu(x) + \mu_0(x)},$$

so our optimal discriminator matches that of Goodfellow et al. (2014) up to a constant factor and logarithm. To recover the minimax formulation, the convex duality (4) yields:

$$\inf_\mu D_{\mathrm{JS}}(\mu, \mu_0) = \inf_\mu \sup_\varphi \mathbb{E}_{x\sim\mu}[\varphi(x)] - \underbrace{\left(-\tfrac{1}{2}\mathbb{E}_{x\sim\mu_0}[\log(1 - e^{2\varphi(x)+\log 2})] - \tfrac{1}{2}\log 2\right)}_{(D_{\mathrm{JS}}(\cdot,\mu_0))^\star(\varphi)}$$

$$= \inf_\mu \sup_D \tfrac{1}{2}\mathbb{E}_{x\sim\mu}[\log(1 - D(x))] + \tfrac{1}{2}\mathbb{E}_{x\sim\mu_0}[\log D(x)],$$

using the substitution $\varphi = \frac{1}{2}\log(1 - D) - \frac{1}{2}\log 2$.

**Non-saturating GAN**  Goodfellow et al. (2014) also proposed the heuristic non-saturating GAN. Theorem 2.5 of Arjovsky & Bottou (2017) shows that the loss function minimized is

$$J(\mu) := D_{\mathrm{KL}}(\tfrac{1}{2}\mu + \tfrac{1}{2}\mu_0 \,\|\, \mu_0) = \frac{1}{2}D_{\mathrm{KL}}(\mu \,\|\, \mu_0) - D_{\mathrm{JS}}(\mu \,\|\, \mu_0).$$

The optimal discriminator is

$$\Phi_\mu(x) = -\frac{1}{2} \log \frac{d\mu_0}{d(\mu + \mu_0)}(x).$$

**Wasserstein GAN**  Arjovsky et al. (2017) proposed the Wasserstein GAN, which minimizes the Wasserstein-1 distance between the input $\mu$ and a fixed measure $\mu_0$:

$$J(\mu) := W_1(\mu, \mu_0) := \inf_\pi \mathbb{E}_{(x,y)\sim\pi}[\|x - y\|],$$

where the infimum is taken over all *couplings* $\pi$, probability distributions over $X \times X$ whose marginals are $\mu$ and $\mu_0$ respectively. The optimal discriminator $\Phi_\mu$ is called the Kantorovich potential in the optimal transport literature (Villani, 2009). The convex duality (4) recover the Wasserstein GAN:

$$\inf_\mu W_1(\mu, \mu_0) = \inf_\mu \sup_\varphi \mathbb{E}_{x\sim\mu}[\varphi(x)] - \underbrace{(\mathbb{E}_{x\sim\mu_0}[\varphi(x)] + \chi\{\|\varphi\|_{\mathrm{Lip}} \leq 1\})}_{(W_1(\cdot,\mu_0))^\star(\varphi)}$$

$$= \inf_\mu \sup_{\|\varphi\|_{\mathrm{Lip}}\leq 1} \mathbb{E}_{x\sim\mu}[\varphi(x)] - \mathbb{E}_{x\sim\mu_0}[\varphi(x)],$$

an expression of Kantorovich–Rubinstein duality. The Lipschitz constraint on the discriminator is typically enforced by spectral normalization (Miyato et al., 2018), less frequently by weight clipping (Arjovsky et al., 2017), or heuristically by gradient penalties (Gulrajani et al., 2017) (although this work shows that gradient penalties may serve a different purpose altogether).

**Maximum mean discrepancy**  Given a positive definite kernel $K : X \times X \to \mathbb{R}$, the maximum mean discrepancy (MMD, Gretton et al. (2012)) between $\mu$ and $\nu$ is defined by

$$J(\mu) := \frac{1}{2}\mathrm{MMD}_K^2(\mu, \nu) := \frac{1}{2} \int K(x, y)\,(\mu - \nu)(dx)\,(\mu - \nu)(dy).$$

where $(\mathcal{H}, \|\cdot\|_{\mathcal{H}})$ is the reproducing kernel Hilbert space (RKHS) for $K$. The generative moment-matching network (GMMN, Li et al. (2015)) and the Coulomb GAN (Unterthiner et al., 2018) use the squared MMD as the loss function. The optimal discriminator in this case is

$$\Phi_\mu(x) = \mathbb{E}_{y\sim\mu}[K(x, y)] - \mathbb{E}_{y\sim\mu_0}[K(x, y)],$$

which in constrast to other GANs, may be approximated by simple Monte Carlo, rather than an auxiliary optimization problem.

Note that MMD-GANs (Li et al., 2017a; Arbel et al., 2018) minimize a modified version of the MMD, the Optimized MMD (Sriperumbudur et al., 2009; Arbel et al., 2018). These MMD-GANs are adversarial in a way that does not arise from convex duality, so our theory currently does not apply to these GANs.

**Integral probability metrics** An integral probability metric (Müller, 1997) is defined by

$$J(\mu) := \mathrm{IPM}_{\mathcal{F}}(\mu, \mu_0) := \sup_{f \in \mathcal{F}} \int f \, d\mu - \int f \, d\mu_0,$$

where $\mathcal{F}$ is a class of functions. The optimal discriminator is the function that maximizes the supremum in the definition. The Wasserstein distance may be thought of as an IPM with $\mathcal{F}$ containing all 1-Lipschitz functions. The MMD may be thought of as an IPM with $\mathcal{F}$ all functions with RKHS norm at most 1, but no GANs based on MMD are actually trained this way, as it is difficult to constrain the discriminator to such functions.

## C   OPTIMAL DISCRIMINATORS ARE FUNCTIONAL DERIVATIVES

Let $J : \mathcal{P}(X) \to \bar{\mathbb{R}}$ be a convex function. Recall the definition of the optimal discriminator (Definition 1):

$$\Phi_\mu \in \arg\max_{\varphi \in \mathcal{C}(X)} \int \varphi \, d\mu - J^\star(\varphi).$$

This definition can be understood as an infinite dimensional analogue of subgradients. In fact, in finite-dimensional convex analysis, $z$ is a subgradient of $f : \mathbb{R}^d \to \bar{\mathbb{R}}$ if and only if it can be written as $z \in \arg\max_{z'} \langle z', x \rangle - f^\star(z')$. The calculus of subgradients shares many properties with the standard calculus of derivatives, such as chain rules (Rockafeller & Wets, 1998). This motivate us to investigate derivative-like features of optimal discriminators.

We introduce the functional derivative, also known as the von Mises influence function:

**Definition 3** (Functional derivative). *Let $J : \mathcal{P}(X) \to \bar{\mathbb{R}}$ be a function of probability measures. We say that a continuous function $\Phi_\mu : X \to \mathbb{R}$ is a **functional derivative** of $J$ at $\mu$ if*

$$J(\mu + \epsilon\xi) = J(\mu) + \epsilon \int \Phi_\mu \, d\xi + O(\epsilon^2)$$

*holds for any $\xi = \nu - \mu$ with $\nu \in \mathcal{P}(X)$.*

Under this definition, optimal discriminators are actually functional derivatives.

**Proposition 16** (Chu et al. (2019), Theorem 2). *Let $J : \mathcal{M}(X) \to \bar{\mathbb{R}}$ be a proper, lower semicontinuous, and convex function. If there exists a maximizer $\Phi_\mu$ of $\varphi \mapsto \int \varphi \, d\mu - J^\star(\varphi)$, then $\Phi_\mu$ is a functional derivative of $J$ at $\mu$ in the sense of Definition 3.*

The following result relates the derivative of the loss function with the derivative of the optimal discriminator:

**Proposition 17** (Chu et al. (2019), Theorem 1). *Let $J : \mathcal{P}(X) \to \mathbb{R}$ be continuously differentiable, in the sense that the functional derivative $\Phi_\mu$ exists and $(\mu, \nu) \mapsto \mathbb{E}_{x \sim \nu}[\Phi_\mu(x)]$ is continuous. Let $\theta \mapsto \mu_\theta$ be continuous in the sense that $\frac{1}{\|h\|}(\mu_{\theta+h} - \mu_\theta)$ converges to a weak limit as $\|h\| \to 0$. Then, we have*

$$\nabla_\theta J(\mu_\theta) = \nabla_\theta \mathbb{E}_{x \sim \mu_\theta}[\hat{\Phi}(x)],$$

*where $\hat{\Phi} = \Phi_{\mu_\theta}$ is treated as a function $X \to \mathbb{R}$ that is not dependent on $\theta$.*

We use this important computational tool in many of our proofs. For the case of the generator model $\mu_\theta = f_{\theta\#}\omega$, an important consequence of Proposition 17 is that

$$\nabla_\theta J(\mu_\theta) = \nabla_\theta \mathbb{E}_{z \sim \omega}[\hat{\Phi}(f_\theta(z))] = \mathbb{E}_{z \sim \omega}[\nabla_{x=f_\theta(z)}\Phi_\mu(f_\theta(z)) \cdot D_\theta f_\theta(z)].$$

We use this fact in the proof of Theorem 1.

## D    PROOFS FOR SECTIONS 1 AND 2

The following result is well known in the dynamical systems and the optimization literature. For the sake of completeness, we provide its proof.

**Proposition 1** (Bertsekas (1999), Proposition 1.2.3). *Suppose $f : \mathbb{R}^p \to \mathbb{R}$ is L-smooth and bounded below. Let $x_{k+1} := x_k - \frac{1}{L}\nabla f(x_k)$. Then $||\nabla f(x_k)|| \to 0$ as $k \to \infty$.*

*Proof.* It is known that if $f$ is $L$-smooth, then

$$f(y) \leq f(x) + \langle \nabla f(x), y - x \rangle + \frac{L}{2}||y - x||^2$$

holds for any $x, y \in \mathbb{R}^d$ (see e.g. Lemma 3.4 in Bubeck (2015)). Then, we have

$$
\begin{aligned}
f(x_{k+1}) &\leq f(x_k) + \langle \nabla f(x_k), x_{k+1} - x_k \rangle + \frac{L}{2}||x_{k+1} - x_k||^2 \\
&\leq f(x_k) - \frac{1}{L}||\nabla f(x_k)||^2 + \frac{1}{2L}||\nabla f(x_k)||^2 \\
&= f(x_k) - \frac{1}{2L}||\nabla f(x_k)||^2.
\end{aligned}
$$

Summing this inequality over $k$, we have

$$\frac{1}{2L}\sum_{k=0}^{n-1}||\nabla f(x_k)||^2 \leq f(x_0) - f(x_n),$$

from which we conclude that

$$\min_{0 \leq k \leq n-1}||\nabla f(x_k)||^2 \leq \frac{2L}{n}(f(x_0) - \inf_x f(x)).$$

$\square$

Next, we move on to prove Theorem 1, which we restate here for readability.

**Theorem 1** (Smoothness decomposition for GANs). *Let $J : \mathcal{M}(X) \to \bar{\mathbb{R}}$ be a convex function whose optimal discriminators $\Phi_\mu : X \to \mathbb{R}$ satisfy the following regularity conditions:*

**(D1)** $x \mapsto \Phi_\mu(x)$ *is $\alpha$-Lipschitz,*

**(D2)** $x \mapsto \nabla_x \Phi_\mu(x)$ *is $\beta_1$-Lipschitz,*

**(D3)** $\mu \mapsto \nabla_x \Phi_\mu(x)$ *is $\beta_2$-Lipschitz w.r.t. the 1-Wasserstein distance.*

*Also, let $\mu_\theta = f_{\theta\#}\omega$ be a family of generators that satisfies:*

**(G1)** $\theta \mapsto f_\theta(z)$ *is $A$-Lipschitz in expectation for $z \sim \omega$, i.e., $\mathbb{E}_{z \sim \omega}[||f_{\theta_1}(z) - f_{\theta_2}(z)||_2] \leq A||\theta_1 - \theta_2||_2$, and*

**(G2)** $\theta \mapsto D_\theta f_\theta(z)$ *is $B$-Lipschitz in expectation for $z \sim \omega$, i.e., $\mathbb{E}_{z \sim \omega}[||D_{\theta_1}f_{\theta_1}(z) - D_{\theta_2}f_{\theta_2}(z)||_2] \leq B||\theta_1 - \theta_2||_2$.*

*Then $\theta \mapsto J(\mu_\theta)$ is L-smooth, with $L = \alpha B + A^2(\beta_1 + \beta_2)$.*

Intuitively, Theorem 1 can be understood as the chain rule. For simplicity, let us consider the smoothness of a composite function $D \circ G$, where $D$ and $G$ are functions on $\mathbb{R}$. A sufficient condition for $D \circ G$ to be smooth is that its second derivative is bounded. For this, suppose that (i) $D$ is $\alpha$-Lipschitz and $\beta$-smooth and (ii) $G$ is $A$-Lipschitz and $B$-smooth. By the chain rule, the second derivative is bounded as $(D \circ G)'' = (D' \circ G)G'' + (D'' \circ G)(G')^2 \leq \alpha B + A^2\beta$, which has the same form as the consequence of Theorem 1. For GANs, we need somewhat more involved conditions; We need two types of smoothness (D2) and (D3) for the optimal discriminators. To this end, we utilize the functional gradient point of view that we explained in Appendix C.

*Proof of Theorem 1.* First, using the functional gradient interpretation of the optimal discriminator, we have

$$
\begin{aligned}
&\|\nabla_{\theta_1} J(\mu_{\theta_1}) - \nabla_{\theta_2} J(\mu_{\theta_2})\| \\
&= \left\|\mathbb{E}_{z\sim\omega}\left[\nabla\Phi_{\mu_{\theta_1}}(f_{\theta_1}(z))\cdot D_{\theta_1}f_{\theta_1} - \nabla\Phi_{\mu_{\theta_2}}(f_{\theta_2}(z))\cdot D_{\theta_2}f_{\theta_2}\right]\right\| \qquad \text{Proposition 17} \\
&= \Big\|\mathbb{E}_{z\sim\omega}\big[\nabla\Phi_{\mu_{\theta_1}}(f_{\theta_1}(z))\cdot(D_{\theta_1}f_{\theta_1} - D_{\theta_2}f_{\theta_2}) \\
&\qquad\qquad + \big(\nabla\Phi_{\mu_{\theta_1}}(f_{\theta_1}(z)) - \nabla\Phi_{\mu_{\theta_2}}(f_{\theta_1}(z))\big)\cdot D_{\theta_2}f_{\theta_2} \\
&\qquad\qquad + \big(\nabla\Phi_{\mu_{\theta_2}}(f_{\theta_1}(z)) - \nabla\Phi_{\mu_{\theta_2}}(f_{\theta_2}(z))\big)\cdot D_{\theta_2}f_{\theta_2}\big]\Big\| \\
&\le \left\|\mathbb{E}_{z\sim\omega}[\nabla\Phi_{\mu_{\theta_1}}(f_{\theta_1}(z))\cdot(D_{\theta_1}f_{\theta_1} - D_{\theta_2}f_{\theta_2})]\right\| && \text{(a)}\\
&\quad + \left\|\mathbb{E}_{z\sim\omega}[(\nabla\Phi_{\mu_{\theta_1}}(f_{\theta_1}(z)) - \nabla\Phi_{\mu_{\theta_2}}(f_{\theta_1}(z)))\cdot D_{\theta_2}f_{\theta_2}]\right\| && \text{(b)}\\
&\quad + \left\|\mathbb{E}_{z\sim\omega}[(\nabla\Phi_{\mu_{\theta_2}}(f_{\theta_1}(z)) - \nabla\Phi_{\mu_{\theta_2}}(f_{\theta_2}(z)))\cdot D_{\theta_2}f_{\theta_2}]\right\|. && \text{(c)}
\end{aligned}
$$

By assumption, there are bounded non-negative numbers $\alpha$, $\beta_1$, $\beta_2$, $A$, and $B$ such that

$$
\begin{aligned}
&\text{(D1}') && \alpha = \sup_{\mu\in\mathcal{P}(X)}\sup_{x\sim\mu}\|\Phi_\mu(x)\|_2, \\
&\text{(D2}') && \sup_{\mu\in\mathcal{P}(X)}\|\nabla_{x_1}\Phi_\mu(x_1) - \nabla_{x_2}\Phi_\mu(x_2)\|_2 \le \beta_1\|x_1 - x_2\|_2 \\
&\text{(D3}') && \sup_{x\in X}\|\nabla_x\Phi_{\mu_1}(x) - \nabla_x\Phi_{\mu_2}(x)\|_2 \le \beta_2 W_1(\mu_1,\mu_2), \\
&\text{(G1}') && A = \mathbb{E}_{z\sim\omega}\sup_\theta\|D_\theta f_\theta(z)\|_{\text{op}}, \quad\text{and} \\
&\text{(G2}') && \mathbb{E}_{z\sim\omega}\|D_{\theta_1}f_{\theta_1}(z) - D_{\theta_2}f_{\theta_2}(z)\|_{\text{op}} \le B\|\theta_1 - \theta_2\|.
\end{aligned}
$$

Here, we wrote $\sup_{x\sim\mu} f(x)$ for the essential supremum of $f$ w.r.t. $\mu$. From (D1$'$) and (G2$'$), the first term (a) is bounded as

$$
\left\|\mathbb{E}_{z\sim\omega}[\nabla\Phi_{\mu_{\theta_1}}(f_{\theta_1}(z))\cdot(D_{\theta_1}f_{\theta_1} - D_{\theta_2}f_{\theta_2})]\right\| \le \alpha B\|\theta_1 - \theta_2\|.
$$

From (D3$'$), (G1$'$), and the Cauchy-Schwarz inequality, the second term (b) is bounded as

$$
\begin{aligned}
&\left\|\mathbb{E}_{z\sim\omega}[(\nabla\Phi_{\mu_{\theta_1}}(f_{\theta_1}(z)) - \nabla\Phi_{\mu_{\theta_2}}(f_{\theta_1}(z)))\cdot D_{\theta_2}f_{\theta_2}]\right\| \\
&\le A\beta_2 W_1(\mu_{\theta_1},\mu_{\theta_2}) \\
&\le A\beta_2\mathbb{E}_{z\sim\omega}\|f_{\theta_1}(z) - f_{\theta_2}(z)\| \le A^2\beta_2\|\theta_1 - \theta_2\|,
\end{aligned}
$$

where the second inequality holds from the following optimal transport interpretation of $W_1$:

$$
W_1(\mu_{\theta_1},\mu_{\theta_2}) = \inf_{\substack{\gamma\in\mathcal{P}(X\times X):\\ \text{coupling of }\mu_{\theta_1}\text{ and }\mu_{\theta_2}}}\int\|x - y\|\,d\gamma \le \mathbb{E}_{z\sim\omega}\|f_{\theta_1}(z) - f_{\theta_2}(z)\|.
$$

Lastly, from (D2$'$), (G1$'$) and the Cauchy-Schwarz inequality, the term (c) is bounded as

$$
\left\|\mathbb{E}_{z\sim\omega}[(\nabla\Phi_{\mu_{\theta_2}}(f_{\theta_1}(z)) - \nabla\Phi_{\mu_{\theta_2}}(f_{\theta_2}(z)))\cdot D_{\theta_2}f_{\theta_2}]\right\| \le A\beta_1\mathbb{E}_{z\sim\omega}\|f_{\theta_1}(z) - f_{\theta_2}(z)\| \le A^2\beta_1\|\theta_1 - \theta_2\|.
$$

Combining the above upper bounds for (a)–(c), we conclude that

$$
\|\nabla_{\theta_1} J(\mu_{\theta_1}) - \nabla_{\theta_2} J(\mu_{\theta_2})\| \le (\alpha B + A^2(\beta_1 + \beta_2))\|\theta_1 - \theta_2\|.
$$

$\square$

# E    PROOFS FOR SECTION 3

In the finite-dimensional case, Proposition 15 says that taking inf-convolution with a "coercive" regularizer does not change the set of minimizers of the original objective. A similar invariance holds for GAN objectives, which are defined on infinite-dimensional space of signed measures, for the regularizers $R_1$, $R_2$ and $R_3$ introduced in Sections 4 to 6:

**Proposition 2** (Invariance of minimizers). *Let $R_1(\xi) := \|\xi\|_{\mathrm{KR}}$, $R_2(\xi) := \|\xi\|_{\mathcal{S}_*}$, and $R_3(\xi) := \frac{1}{2}\|\hat{\xi}\|_{\mathcal{H}}^2$ be the three regularizers defined by (8), (12), and (19) respectively. Assume that $J : \mathcal{M}(X) \to \bar{\mathbb{R}}$ has a unique minimizer at $\mu_0$ with $J(\mu_0) = 0$, and $J(\mu) \geq c\|\hat{\mu} - \hat{\mu}_0\|_{\mathcal{H}}$ for some $c > 0$. Then the inf-convolution $\tilde{J} = J \oplus R_1 \oplus R_2 \oplus R_3$ has a unique minimizer at $\mu_0$ with $\tilde{J}(\mu_0) = 0$.*

*Proof.* In the following, we endow $\mathcal{M}(X)$ with the Gaussian RKHS norm $\|\hat{\cdot}\|_{\mathcal{H}}$, and for convenience $\|\cdot\|$ will refer to this norm if not otherwise specified. To show the result, we apply Lemma 1 three times, first on $R_3$ and $R_1$, and then again on $R_3 \oplus R_1$ and $R_2$, and then finally on $R_3 \oplus R_1 \oplus R_2$ and $J(\cdot + \mu_0)$. The result follows from noting that $\tilde{J}(\mu) = [R_3 \oplus R_1 \oplus R_2 \oplus J(\cdot + \mu_0)](\mu - \mu_0)$.

In order to apply the lemma, it suffices to show that $R_3$ is uniformly continuous on bounded sets and coercive, and that there exist constants $c_1$ and $c_2$ such that $R_1(\xi) \geq c_1\|\xi\|$ and $R_2(\xi) \geq c_2\|\xi\|$. $R_3$ is uniformly continuous on bounded sets: suppose $\|\xi\| < C$ and $\|\xi'\| < C$; then $|R_3(\xi) - R_3(\xi')| = |\frac{1}{2}\|\xi\|^2 - \frac{1}{2}\|\xi'\|^2| = |\frac{1}{2}\langle \xi - \xi', \xi + \xi'\rangle| \leq C\|\xi - \xi'\|$ by the Cauchy–Schwarz and triangle inequality. $R_3$ is also coercive, as $R_3(\xi) = \frac{1}{2}\|\xi\|^2 \to \infty$ when $\|\xi\| \to \infty$. $R_1$ satisfies $R_1(\xi) \geq c_1\|\xi\|$ by Lemma 7. $R_2$ satisfies $R_2(\xi) \geq c_2\|\xi\|$ by Lemma 5. □

Recall that a function $F$ is said to be *coercive* if $F(\xi) \to \infty$ when $\|\xi\| \to \infty$.

**Lemma 1.** *Suppose $F : \mathcal{M}(X) \to \mathbb{R}$ has a unique minimizer at $0$ with $F(0) = 0$, and is uniformly continuous on bounded sets and coercive. Suppose $G : \mathcal{M}(X) \to \bar{\mathbb{R}}$ has a unique minimizer at $0$ with $G(0) = 0$, and $G(\cdot) \geq c\|\cdot\|$ for some $c > 0$. Then the inf-convolution $F \oplus G$ has a unique minimizer at $0$ with $(F \oplus G)(0) = 0$, and is uniformly continuous on bounded sets, coercive, and real-valued.*

*Proof.* From Theorem 2.3 of Strömberg (1996), $\inf F \oplus G = 0$, and $0 \in \arg\min F \oplus G$. To show that $0$ is the unique minimizer, let $\xi \in \arg\min F \oplus G$. From the definition of inf-convolution, for every $n$ there exists an $\xi_n \in \mathcal{M}(X)$ that satisfies

$$\lim_{n \to \infty} F(\xi_n) + G(\xi - \xi_n) = (F \oplus G)(\xi) = 0.$$

Since $F$ and $G$ are non-negative, $F(\xi_n) \to 0$ and $G(\xi - \xi_n) \to 0$. By our assumption that $G(\cdot) \geq c\|\cdot\|$, the latter limit implies that $\|\xi - \xi_n\| \to 0$, which implies that $F(\xi_n) \to F(\xi)$ by continuity of $F$. Comparing our two expressions for the limit of $F(\xi_n)$, we find that $F(\xi) = 0$, which implies that $\xi = 0$, since $F$ has a unique minimizer at $0$. Hence $\arg\min F \oplus G = \{0\}$.

From Theorem 2.10 of Strömberg (1996), $F \oplus G$ is uniformly continuous on bounded sets and real-valued. To show that $F \oplus G$ is coercive, we show the equivalent condition its sublevel sets are bounded (see Proposition 11.11 of Bauschke & Combettes (2011)). That is, we show that for all $a > 0$, there exists a constant $b$ such that if $(F \oplus G)(\xi) \leq a$, then $\|\xi\| \leq b$. Let $a > 0$, and let $\xi \in \mathcal{M}(X)$ be such that $(F \oplus G)(\xi) \leq a$. Let $\epsilon > 0$; from the definition of inf-convolution, there exists a $\tilde{\xi} \in \mathcal{M}(X)$ such that

$$F(\tilde{\xi}) + G(\xi - \tilde{\xi}) \leq (F \oplus G)(\xi) + \epsilon \leq a + \epsilon.$$

Because $F$ and $G$ are non-negative, we have that

$$F(\tilde{\xi}) \leq a + \epsilon, \qquad G(\xi - \tilde{\xi}) \leq a + \epsilon.$$

Using the fact that $F$ is coercive and hence its sublevel sets are bounded, let $b'$ be such that if $F(\nu) \leq a + \epsilon$, then $\|\nu\| \leq b'$. By the triangle inequality and our assumption that $G(\cdot) \geq c\|\cdot\|$,

$$\|\xi\| \leq \|\tilde{\xi}\| + \|\xi - \tilde{\xi}\| \leq b' + \frac{G(\xi - \tilde{\xi})}{c} \leq b' + \frac{a + \epsilon}{c}.$$

Because this constant is independent of $\xi$, this shows that $F \oplus G$ is coercive.

□

# F PROOFS FOR SECTION 4

**Proposition 3.** *(D1) holds if and only if $J$ is $\alpha$-Lipschitz w.r.t. the Wasserstein-1 distance.*

*Proof.* First, we check the forward implication:

$$|J(\mu) - J(\nu)| \le \alpha W_1(\mu, \nu) \quad \Rightarrow \quad \sup_{x \sim \mu} \|\nabla_x \Phi_\mu(x)\|_2 \le \alpha.$$

From the duality between $L^\infty$- and $L^1$-norms, we have

$$\sup_{x \sim \mu} \|\nabla_x \Phi_\mu(x)\|_2 = \|\nabla_x \Phi_\mu(\cdot)\|_{L^\infty(\mu; \mathbb{R}^d)} = \sup_{\|v\|_{L^1(\mu; \mathbb{R}^d)}=1} \int \nabla \Phi_\mu(x) \cdot v(x) \, d\mu(x). \quad (26)$$

Below, we will abbreviate $\|\cdot\|_{L^1(\mu; \mathbb{R}^d)}$ as $\|\cdot\|_{L^1}$. We utilize the fact that the optimal discriminator $\Phi_\mu$ is the functional gradient of $J$ in the sense of Definition 3. For any $t \ge 0$, let $\mu_t = (\mathrm{id} + tv)_\# \mu$ be the law of $x + tv(x)$ with $x \sim \mu$. Then, $\mu_t$ converges weakly to $\mu = \mu_0$ as $t \to 0$ since $W_1(\mu_t, \mu) \le t\|v\|_{L^1(\mu)} \to 0$. Applying Proposition 17 to $J$ and $\mu_t$, we have

$$\frac{d}{dt} J(\mu_t)\Big|_{t=0} = \frac{d}{dt} \int \Phi_\mu(x + tv(x)) \, d\mu(x)\Big|_{t=0} = \int \nabla \Phi_\mu(x) \cdot v(x) \, d\mu.$$

In particular, the right-hand side of (26) is bounded from above as

$$\sup_{\|v\|_{L^1(\mu)}=1} \int \nabla \Phi_\mu(x) \cdot v(x) \, d\mu(x)$$

$$= \sup_{\|v\|_{L^1(\mu)}=1} \frac{d}{dt} J(\mu_t)\Big|_{t=0}$$

$$= \sup_{\|v\|_{L^1(\mu)}=1} \lim_{t \to 0} \frac{J(\mu_t) - J(\mu)}{t}$$

$$\le \sup_{\|v\|_{L^1(\mu)}=1} \lim_{t \to 0} \frac{\alpha W_1(\mu_t, \mu)}{t}$$

$$\le \sup_{\|v\|_{L^1(\mu)}=1} \lim_{t \to 0} \frac{\alpha t \|v\|_{L^1(\mu)}}{t}$$

$$= \alpha,$$

which proves the first half of the statement.

For the converse, we borrow some techniques from the optimal transport theory. Let $p > 1$, and let $W_p(\mu, \nu)$ denote the Wasserstein-$p$ distance between $\mu$ and $\nu$. Suppose that $\mu_t : [0, 1] \to \mathcal{P}(X)$ is an $\mathcal{P}_p(X)$-absolutely continuous curve, that is, every $\mu_t$ has a finite $p$-moment and there exists $v \in L^1([0, 1])$ such that $W_p(\mu_s, \mu_t) \le \int_s^t v(r) \, dr$ holds for any $0 \le s \le t \le 1$. Then, the limit $|\mu'|_{W_p}(t) := \lim_{h \to 0} |h|^{-1} W_p(\mu_{t+h}, \mu_t)$ exists for almost all $t \in [0, 1]$ (see Ambrosio et al. (2008), Theorem 1.1.2). Such $|\mu'|_{W_p}$ is called the metric derivative of $\mu_t$.

**Lemma 2** (Ambrosio et al. (2008), Theorem 8.3.1). *Let $\mu_t : [0, 1] \to \mathcal{P}(X)$ be an $\mathcal{P}_p(X)$-absolutely continuous curve, and let $|\mu'|_{W_p} \in L^1([0, 1])$ be its metric derivative. Then, there exists a vector field $v : (x, t) \mapsto v_t(x) \in \mathbb{R}^d$ such that*

$$v_t \in L^p(\mu_t; X), \quad \|v_t\| \le |\mu'|_{W_p}(t)$$

*for almost all $t \in [0, 1]$, and*

$$\int_0^1 \frac{d}{dt} \varphi(x, t) \mu_t(dx) dt = -\int_0^1 \langle v_t(x), \nabla_x \varphi(x, t) \rangle \mu_t(dx) dt$$

*holds for any cylindrical function $\varphi(x, t)$.*

Given $p > 1$, let $\mu_t$ be a $\mathcal{P}_p(X)$-absolutely continuous curve such that $\mu_0 = \mu$ and $\mu_1 = \nu$. Then

$$
\begin{aligned}
|J(\mu) - J(\nu)| &= \left| \int_0^1 \frac{d}{dt} J(\mu_t) \, dt \right| \\
&= \left| \int_0^1 \int v_t \cdot \nabla \Phi_{\mu_t} \, d\mu_t \, dt \right| \\
&\leq \int_0^1 \|v_t\|_{L^p(\mu_t)} \|\nabla \Phi_{\mu_t}\|_{L^q(\mu_t)} \, dt \\
&\leq \int_0^1 \|v_t\|_{L^p(\mu_t)} \, dt \cdot \sup_{\tilde{\mu}} \|\nabla \Phi_{\tilde{\mu}}\|_{L^q(\mu)} \\
&\leq \int_0^1 |\mu_t'|_{W_p} \, dt \cdot \sup_{\tilde{\mu}} \|\nabla \Phi_{\tilde{\mu}}\|_{L^q(\mu)},
\end{aligned}
$$

where $|\mu_t'|$ is the metric derivative. The existence of $v_t$ and the bound $\|v_t\|_{L^p(\mu_t)} \leq |\mu_t'|_{W_p}$ is due to Lemma 2. Taking the infimum over all possible such curves, we find that

$$
|J(\mu) - J(\nu)| \leq W_p(\mu, \nu) \cdot \sup_{\tilde{\mu}} \|\nabla \Phi_{\tilde{\mu}}\|_{L^q(\mu)}.
$$

To conclude, we consider the limit $p \to 1$. Using the fact that $q \mapsto \|f\|_{L^q(\mu)}$ is increasing in $q$, we have

$$
|J(\mu) - J(\nu)| \leq W_p(\mu, \nu) \cdot \sup_{\tilde{\mu}} \|\nabla \Phi_{\tilde{\mu}}\|_{L^\infty(\mu)} \leq \alpha W_p(\mu, \nu).
$$

Then

$$
\begin{aligned}
|J(\mu) - J(\nu)| &\leq \inf_{p > 1} \alpha W_p(\mu, \nu) \\
&= \alpha \inf_{\pi \in \Pi(\mu, \nu)} \inf_{p > 1} \|x - y\|_{L^p(\pi)} \\
&= \alpha \inf_{\pi \in \Pi(\mu, \nu)} \|x - y\|_{L^1(\pi)} \\
&= \alpha W_1(\mu, \nu).
\end{aligned}
$$

$\square$

**Proposition 4.** *The minimax and non-saturating GAN losses do not satisfy (D1) for some $\mu_0$.*

**Proposition 5.** *The Wasserstein GAN loss satisfies (D1) with $\alpha = 1$ for any $\mu_0$.*

*Proof for Proposition 4 and Proposition 5.* The counterexample for Proposition 4 is a slight simplification of Example 1 of Arjovsky et al. (2017). Let $\delta_x$ denote the distribution that outputs $x \in \mathbb{R}$ with probability 1. Then, we have

$$
D_{\mathrm{JS}}(\delta_x \,\|\, \delta_0) = \begin{cases} \log 2 & x \neq 0 \\ 0 & x = 0, \end{cases} \quad D_{\mathrm{KL}}(\tfrac{1}{2}\delta_x + \tfrac{1}{2}\delta_0 \,\|\, \delta_0) = \begin{cases} \infty & x \neq 0 \\ 0 & x = 0, \end{cases} \quad \text{and} \quad W_1(\delta_x, \delta_0) = |x|.
$$

Therefore, for $J(\mu) = D_{\mathrm{JS}}(\mu \,\|\, \delta_0)$, the inequality $|J(\delta_x) - J(\delta_0)| \leq \alpha W_1(\delta_x, \delta_0)$ cannot hold for sufficiently small $x \neq 0$, because the left-hand side equals $\log 2$ while the right-hand side equals $\alpha|x|$. For $J(\mu) = D_{\mathrm{KL}}(\tfrac{1}{2}\mu + \tfrac{1}{2}\delta_0 \,\|\, \delta_0)$, the inequality will not hold for any $x \neq 0$.

Next, we verify Proposition 5. For $J(\mu) = W_1(\mu, \mu_0)$, we have $J(\mu) - J(\nu) = W_1(\mu, \mu_0) - W_1(\nu, \mu_0) \leq W_1(\mu, \nu)$ by the triangle inequality (see e.g., Section 6 in Villani (2009)). Reversing the roles of $\mu$ and $\nu$, we can conclude that $|J(\mu) - J(\nu)| \leq W_1(\mu, \nu)$. The result follows from Proposition 3. $\square$

**Proposition 6** (Pasch–Hausdorff). *Let $J : \mathcal{M}(X) \to \bar{\mathbb{R}}$ be a function, and define $\tilde{J} := J \oplus R_1$. Then $\tilde{J}$ is $\alpha$-Lipschitz w.r.t. the distance induced by the KR norm, and hence the Wasserstein-1 distance when restricted to $\mathcal{P}(X)$.*

*Proof.* Observe that by the triangle inequality,

$$
\begin{aligned}
\tilde{J}(\mu) - \tilde{J}(\nu) &= \sup_{\tilde{\nu}} \inf_{\tilde{\mu}} J(\tilde{\mu}) + \alpha W_1(\mu, \tilde{\mu}) - J(\tilde{\nu}) - \alpha W_1(\nu, \tilde{\nu}) \\
&\leq \sup_{\tilde{\nu}} \alpha W_1(\mu, \tilde{\nu}) - \alpha W_1(\nu, \tilde{\nu}) \\
&\leq \sup_{\tilde{\nu}} \alpha W_1(\mu, \nu) \\
&= \alpha W_1(\mu, \nu).
\end{aligned}
$$

Interchanging the roles of $\mu$ and $\nu$ completes the proof. $\qquad\square$

**Lemma 3.** *The convex conjugate of $\mu \mapsto \alpha ||\mu||_{\mathrm{KR}}$ is given as $\varphi \mapsto \chi\{||\varphi||_{\mathrm{Lip}} \leq \alpha\}$.*

*Proof.* The convex conjugate is

$$
\begin{aligned}
\sup_{\mu} \Big[ \int f \, d\mu - \alpha ||\mu||_{\mathrm{KR}} \Big] &= \sup_{\mu} \Big[ \int f \, d\mu - \alpha \sup_{||g||_{\mathrm{Lip}} \leq 1} \int g \, d\mu \Big] \\
&= \sup_{\mu} \Big[ \int f \, d\mu - \sup_{||g||_{\mathrm{Lip}} \leq \alpha} \int g \, d\mu \Big] \\
&= \sup_{\mu} \Big[ \inf_{||g||_{\mathrm{Lip}} \leq \alpha} \int f \, d\mu - \int g \, d\mu \Big] \\
&= \inf_{||g||_{\mathrm{Lip}} \leq \alpha} \sup_{\mu} \Big[ \int f \, d\mu - \int g \, d\mu \Big] \\
&= \inf_{||g||_{\mathrm{Lip}} \leq \alpha} \chi\{f = g\} \\
&= \chi\{||f||_{\mathrm{Lip}} \leq \alpha\},
\end{aligned}
$$

where Sion's minimax theorem ensures that we can swap the inf and the sup. $\qquad\square$

## G   PROOFS FOR SECTION 5

**Proposition 7.** *The Wasserstein, minimax, and non-saturating GAN losses do not satisfy (D2) for some $\mu_0$.*

*Proof.* First, we prove the statement for the Wasserstein GAN. Consider $J(\mu) = W_1(\mu, \delta_0)$ evaluated at $\mu = \frac{1}{2}\delta_{-1} + \frac{1}{2}\delta_1$, a mixture of spikes at $\pm 1$. The optimal discriminator of $J$ at this mixture is the Kantorovich potential that transfers $\mu$ to $\delta_0$, which is $\Phi_\mu(x) = |x|$. The gradient of this Kantorovich potential is discontinuous at 0, and hence not Lipschitz.

For the minimax GAN $J(\mu) = D_{\mathrm{JS}}(\mu \,||\, \mu_0)$, let $\mu$ and $\mu_0$ be probability distributions on $\mathbb{R}$ whose densities are $p(x) \propto \exp(-|x|/2)$ and $p_0(x) \propto \exp(-|x|)$, respectively (i.e., Laplace distributions). Then, the optimal discriminator at $\mu$ is given as $\Phi_\mu(x) = \frac{1}{2} \log \frac{p(x)}{p(x) + p_0(x)} = -\frac{1}{2} \log(1 + 2\exp(-|x|/2))$. By elementary calculations, we can see that this function is not differentiable at $x = 0$, which implies that the minimax GAN does not satisfy (D2). For the non-saturating GAN, we obtain a similar result by swapping the role of $\mu$ and $\mu_0$. $\qquad\square$

**Proposition 8.** *Let $J : \mathcal{M}(X) \to \bar{\mathbb{R}}$ be a convex, proper, lower semicontinuous function, and define $\tilde{J} := J \oplus R_2$. Then the optimal discriminator for $\tilde{J}$ is $\beta_1$-smooth.*

*Proof.* $\tilde{J}$ is convex, proper, lower semicontinuous. Hence

$$
\tilde{J}(\mu) = (J \oplus \lambda || \cdot ||_{\mathcal{F}^*})(\mu) = \sup_{\varphi} \int \varphi \, d\mu - J^\star(\varphi) - \chi\{\tfrac{1}{\lambda}\varphi \in \mathcal{F}\}.
$$

Then by the envelope theorem, $\frac{\delta J}{\delta \mu}$ is the $\varphi$ that maximizes the right-hand side, and hence $\frac{1}{\lambda} \frac{\delta J}{\delta \mu} \in \mathcal{F}$. $\qquad\square$

**Lemma 4.** *The convex conjugate of $\mu \mapsto \beta_1 \|\mu\|_{\mathcal{S}^*}$ is $\varphi \mapsto \chi\{\varphi \in \beta_1 \mathcal{S}\}$.*

*Proof.* The proof of Lemma 4 is nearly identical to the proof of Lemma 3. $\qquad \square$

**Proposition 9.** *Let $f : \mathbb{R}^d \to \mathbb{R}$ be a neural network consisting of $k$ layers whose linear transformations have spectral norm $1$ and whose activation functions are $1$-Lipschitz and $1$-smooth. Then $f$ is $k$-smooth.*

*Proof.* In this section, let $\ell(f)$ and $s(f)$ denote the Lipschitz constant of $f$ and $\nabla f$ respectively. This inequality holds

$$s(f \circ g) \leq s(f)\ell(g)^2 + \ell(f)s(g), \tag{27}$$

since

$$
\begin{aligned}
\|d(f \circ g)(x) - d(f \circ g)(y)\| &= \|df(g(x))dg(x) - df(g(y))dg(y)\| \\
&= \|df(g(x))dg(x) - df(g(y))dg(x) + df(g(y))dg(x) - df(g(y))dg(y)\| \\
&\leq \|df(g(x)) - df(g(y))\| \, \|dg(x)\| + \|df(g(y))\| \, \|dg(x) - dg(y)\| \\
&\leq s(f)\|g(x) - g(y)\|\ell(g) + \ell(f)s(g)\|x - y\| \\
&\leq \big(s(f)\ell(g)^2 + \ell(f)s(g)\big)\|x - y\|.
\end{aligned}
$$

Let $\sigma$ be an elementwise activation function with $\ell(\sigma) = s(\sigma) = 1$, and let $A$ be a linear layer with spectral norm $1$. Then $\ell(A) = 1$ and $s(A) = 0$, so

$$
\ell(\sigma \circ A) \leq \ell(\sigma)\,\ell(A) = 1
$$
$$
s(\sigma \circ A) \leq s(\sigma)\ell(A)^2 + \ell(\sigma)s(A) = 1.
$$

We note that we can use the same value of $s(\sigma)$ whether we consider $\sigma : \mathbb{R} \to \mathbb{R}$ or elementwise as $\sigma : \mathbb{R}^d \to \mathbb{R}^d$, since for an elementwise $\sigma$, we have

$$
\begin{aligned}
\|d\sigma(x) - d\sigma(y)\|_2 &= \|\operatorname{diag}\sigma'(x) - \operatorname{diag}\sigma'(y)\|_2 \\
&= \|\operatorname{diag}(\sigma'(x) - \sigma'(y))\|_2 \\
&= \max_i |\sigma'(x_i) - \sigma'(y_i)| \\
&\leq \max_i s(\sigma)|x_i - y_i| \\
&= s(\sigma)\|x - y\|_\infty \\
&\leq s(\sigma)\|x - y\|_2.
\end{aligned}
$$

We apply the inequality (27) recursively for all $k$ layers to obtain that the entire network is $k$-smooth. $\qquad \square$

**Lemma 5.** *Let $\mathcal{H}$ be an RKHS with the Gaussian kernel. Then $\|\hat{\mu}\|_{\mathcal{H}} \leq \frac{\sqrt{2}}{\sigma^2}\|\mu\|_{\mathcal{S}^*}$.*

*Proof.* It follows from $\frac{\partial f(x)}{\partial x_i} = \langle f, \frac{\partial K(x,\cdot)}{\partial x_i}\rangle$ that, for $f \in \mathcal{H}$,

$$
\begin{aligned}
\|\nabla f(x) - \nabla f(y)\|^2 &= \sum_{i=1}^d \left|\frac{\partial f(x)}{\partial x_i} - \frac{\partial f(y)}{\partial y_i}\right|^2 \\
&\leq \sum_{i=1}^d \|f\|_{\mathcal{H}}^2 \left\|\frac{\partial K(x,\cdot)}{\partial x_i} - \frac{\partial K(y,\cdot)}{\partial y_i}\right\|_{\mathcal{H}}^2 \\
&= \|f\|_{\mathcal{H}}^2 \sum_{i=1}^d \left\{\frac{\partial^2 K(x,\tilde{x})}{\partial x_i \partial \tilde{x}_i} - 2\frac{\partial^2 K(x,y)}{\partial x_i \partial y_i} + \frac{\partial^2 K(y,\tilde{y})}{\partial y_i \partial \tilde{y}_i}\right\} \\
&\leq \|f\|_{\mathcal{H}}^2 \cdot (\Gamma(x,y) + \Gamma(y,x)),
\end{aligned}
$$

where $\tilde{x}$ and $\tilde{y}$ denote copies of $x$ and $y$, and

$$
\Gamma(x,y) = \sum_{i=1}^d \left|\frac{\partial^2 K(x,\tilde{x})}{\partial x_i \partial \tilde{x}_i} - \frac{\partial^2 K(x,y)}{\partial x_i \partial \tilde{x}_i}\right|.
$$

From (30), we find that for a Gaussian kernel,

$$\Gamma(x, y) = \frac{1}{\sigma^4} \exp\left(-\frac{\|x - y\|^2}{2\sigma^2}\right) \|x - y\|^2,$$

and thus,

$$\|\nabla f(x) - \nabla f(y)\| \le \frac{\sqrt{2}}{\sigma^2} \|f\|_{\mathcal{H}} \|x - y\|.$$

This implies that

$$\|\hat{\mu}\|_{\mathcal{H}} = \sup_{f \in \mathcal{H}, \|f\|_{\mathcal{H}} \le 1} \int f \, d\mu \le \sup_{\|f\|_{\mathcal{S}} \le \frac{\sqrt{2}}{\sigma^2}} \int f \, d\mu = \frac{\sqrt{2}}{\sigma^2} \|\mu\|_{\mathcal{S}^*}.$$

$\square$

## H  PROOFS FOR SECTION 6

**Lemma 6.** *The convex conjugate of* $\mu \mapsto \frac{\lambda}{2} \|\mu\|_{\mathrm{KR}}^2$ *is* $f \mapsto \frac{1}{2\lambda} \|f\|_{\mathrm{Lip}}^2$.

*Proof.* This proof is generalized from the finite-dimensional case (Boyd & Vandenberghe, 2004). We can bound the convex conjugate from above by

$$\sup_{\mu} \left[ \int f \, d\mu - \frac{\lambda}{2} \|\mu\|_{\mathrm{KR}}^2 \right] \le \sup_{\mu} \left[ \|f\|_{\mathrm{Lip}} \|\mu\|_{\mathrm{KR}} - \frac{\lambda}{2} \|\mu\|_{\mathrm{KR}}^2 \right]$$

$$\le \sup_{z \in \mathbb{R}} \left[ \|f\|_{\mathrm{Lip}} z - \frac{\lambda}{2} z^2 \right]$$

$$= \|f\|_{\mathrm{Lip}} \cdot \frac{\|f\|_{\mathrm{Lip}}}{\lambda} - \frac{\lambda}{2} \cdot \frac{\|f\|_{\mathrm{Lip}}^2}{\lambda^2}$$

$$= \frac{1}{2\lambda} \|f\|_{\mathrm{Lip}}^2.$$

If $f$ is constant, then we may choose $\mu = 0$ to see that

$$\sup_{\mu} \left[ \int f \, d\mu - \frac{\lambda}{2} \|\mu\|_{\mathrm{KR}}^2 \right] \ge 0 = \frac{1}{2\lambda} \|f\|_{\mathrm{Lip}}^2,$$

and we are done.

Otherwise, for $f(x) \ne f(y)$, let $\mu_{x,y}$ be the signed measure given by

$$\mu_{x,y} = \frac{\|f\|_{\mathrm{Lip}}^2}{\lambda(f(x) - f(y))} (\delta_x - \delta_y).$$

Note that

$$\|\mu_{x,y}\|_{\mathrm{KR}} = \sup_{\|g\|_{\mathrm{Lip}} \le 1} \int g \, d\mu_{x,y}$$

$$= \frac{\|f\|_{\mathrm{Lip}}^2}{\lambda(f(x) - f(y))} \sup_{\|g\|_{\mathrm{Lip}} \le 1} \left( g(x) - g(y) \right)$$

$$= \frac{\|f\|_{\mathrm{Lip}}^2}{\lambda(f(x) - f(y))} \cdot d(x, y)$$

and

$$\int f \, d\mu_{x,y} = \frac{\|f\|_{\mathrm{Lip}}^2}{\lambda(f(x) - f(y))} \left( f(x) - f(y) \right) = \frac{1}{\lambda} \|f\|_{\mathrm{Lip}}^2.$$

Then

$$\sup_{\mu} \Big[ \int f \, d\mu - \frac{\lambda}{2} ||\mu||^2_{\mathrm{KR}} \Big] \geq \sup_{f(x) \neq f(y)} \Big[ \int f \, d\mu_{x,y} - \frac{\lambda}{2} ||\mu_{x,y}||^2_{\mathrm{KR}} \Big]$$

$$= \sup_{f(x) \neq f(y)} \Big[ \frac{1}{\lambda} ||f||^2_{\mathrm{Lip}} - \frac{\lambda}{2} \Big( \frac{1}{\lambda} ||f||^2_{\mathrm{Lip}} \frac{d(x,y)}{f(x) - f(y)} \Big)^2 \Big]$$

$$= \frac{1}{\lambda} ||f||^2_{\mathrm{Lip}} - \frac{\lambda}{2} \Big( \frac{1}{\lambda} ||f||^2_{\mathrm{Lip}} \frac{1}{||f||_{\mathrm{Lip}}} \Big)^2$$

$$= \frac{1}{2\lambda} ||f||^2_{\mathrm{Lip}}.$$

$\square$

**Proposition 10.** *Let $J : \mathcal{M}(X) \to \mathbb{R}$ be a convex function. Then $J$ satisfies (D3) if and only if $\mathfrak{D}_J(\nu, \mu) \leq \frac{\beta_2}{2} ||\mu - \nu||^2_{\mathrm{KR}}$ for all $\mu, \nu \in \mathcal{M}(X)$.*

*Proof.* Recall the definition of the Bregman divergence:

$$\mathfrak{D}_J(\nu, \mu) := J(\nu) - J(\mu) - \int \Phi_\mu(x) \, d(\nu - \mu).$$

Proposition 10 claims that the optimal discriminator $\Phi_\mu$ is $\beta_2$-smooth as a function of $\mu$ if and only if

$$\forall \mu, \nu \in \mathcal{M}(X), \quad J(\nu) - J(\mu) - \int \Phi_\mu(x) \, d(\nu - \mu) \leq \frac{\beta_2}{2} ||\mu - \nu||^2_{\mathrm{KR}}. \tag{28}$$

This proof is generalized from the finite-dimensional case (Zhou, 2018; Sidford, 2017). Suppose that $||\nabla \Phi_\mu(x) - \nabla \Phi_\nu(x)||_2 \leq \beta_2 ||\mu - \nu||_{\mathrm{KR}}$ holds for all $x$. Let $\mu_t$ ($0 \leq t \leq 1$) be defined as a mixture between $\mu_t = (1 - t)\mu + t\nu$ so that

$$\mu_0 = \mu, \quad \mu_1 = \nu, \quad \text{and} \quad ||\mu_t - \mu_s||_{\mathrm{KR}} = |t - s| \cdot ||\mu - \nu||_{\mathrm{KR}}.$$

Then, we have

$$|\mathfrak{D}_J(\nu, \mu)| = \Big| \int_0^1 \frac{d}{dt} J(\mu_t) \, dt - \int \Phi_\mu \, d(\nu - \mu) \Big|$$

$$= \Big| \int_0^1 \int \Phi_{\mu_t} \, d(\nu - \mu) \, dt - \int \Phi_\mu \, d(\nu - \mu) \Big|$$

$$= \Big| \int_0^1 \int (\Phi_{\mu_t} - \Phi_\mu) \, d(\nu - \mu) \, dt \Big|$$

$$\leq \int_0^1 ||\Phi_{\mu_t} - \Phi_\mu||_{\mathrm{Lip}} ||\mu - \nu||_{\mathrm{KR}} \, dt$$

$$\leq \int_0^1 \beta_2 ||\mu_t - \mu||_{\mathrm{KR}} ||\mu - \nu||_{\mathrm{KR}} \, dt$$

$$\leq \int_0^1 t\beta_2 ||\mu - \nu||^2_{\mathrm{KR}} \, dt$$

$$= \frac{\beta_2}{2} ||\mu - \nu||^2_{\mathrm{KR}}.$$

Since the choice of $\mu$ and $\nu$ is arbitrary, we have proved (28) by assuming (D2).

Next, we move on to prove the converse. Before proceeding, note that the Bregman divergence $\mathfrak{D}_J(\nu, \mu)$ is always non-negative. In fact, for any $\mu, \nu \in \mathcal{M}(X)$, we have

$$\int \Phi_\mu \, d(\nu - \mu) = \underbrace{\int \Phi_\mu \, d\nu - J^\star(\Phi_\mu)}_{\leq J(\nu)} - \underbrace{\Big[ \int \Phi_\mu \, d\mu - J^\star(\Phi_\mu) \Big]}_{=J(\mu)}$$

$$\leq J(\nu) - J(\mu),$$

which implies $\mathfrak{D}_J(\nu, \mu) \geq 0$.

Choose $\xi \in \mathcal{M}(X)$ arbitrarily. If $\xi = \mu$, the inequality $\|\nabla \Phi_\mu - \nabla \Phi_\xi\|_2 \leq \beta_2 \|\xi - \mu\|_{\mathrm{KR}}$ is trivial, so we assume $\xi \neq \mu$ below. By the non-negativity of the Bregman divergence, we have

$$
\begin{aligned}
0 \leq \ &\mathfrak{D}_J(\nu, \mu) \\
= \ &J(\nu) - \left[ J(\mu) + \int \Phi_\mu \, d(\nu - \mu) \right] \\
\leq \ &\left[ J(\xi) + \int \Phi_\xi \, d(\nu - \xi) + \frac{\beta_2}{2} \|\xi - \nu\|_{\mathrm{KR}}^2 \right] - \left[ J(\mu) + \int \Phi_\mu \, d(\nu - \mu) \right] \\
= \ &J(\xi) - J(\mu) - \int \Phi_\mu \, d(\xi - \mu) + \int [\Phi_\xi - \Phi_\mu] \, d(\nu - \xi) + \frac{\beta_2}{2} \|\xi - \nu\|_{\mathrm{KR}}^2. \quad (29)
\end{aligned}
$$

Since the above inequality holds for all $\nu \in \mathcal{M}(X)$, the last expression is still non-negative if we take the infimum over all $\nu$. In particular,

$$
\begin{aligned}
\inf_{\nu \in \mathcal{M}(X)} &\int [\Phi_\xi - \Phi_\mu] \, d(\nu - \xi) + \frac{\beta_2}{2} \|\xi - \nu\|_{\mathrm{KR}}^2 \\
&= - \sup_{\nu \in \mathcal{M}(X)} \left[ \int [\Phi_\mu - \Phi_\xi] \, d(\nu - \xi) - \frac{\beta_2}{2} \|\xi - \nu\|_{\mathrm{KR}}^2 \right] \\
&= - \sup_{\zeta \in \mathcal{M}(X)} \left[ \int [\Phi_\mu - \Phi_\xi] \, d\zeta - \frac{\beta_2}{2} \|\zeta\|_{\mathrm{KR}}^2 \right] \\
&= - \frac{1}{2\beta_2} \|\Phi_\mu - \Phi_\xi\|_{\mathrm{Lip}}^2,
\end{aligned}
$$

using Lemma 6 for the last equality.

Continuing from (29), we have

$$
0 \leq J(\xi) - J(\mu) - \int \Phi_\mu \, d(\xi - \mu) - \frac{1}{2\beta_2} \|\Phi_\mu - \Phi_\xi\|_{\mathrm{Lip}}^2.
$$

Swapping the roles of $\mu$ and $\xi$, we obtain a similar inequality. Adding both sides of thus obtained two inequalities, we obtain

$$
0 \leq \int [\Phi_\xi - \Phi_\mu] \, d(\xi - \mu) - \frac{1}{\beta_2} \|\Phi_\mu - \Phi_\xi\|_{\mathrm{Lip}}^2.
$$

Finally, we have

$$
\|\Phi_\mu - \Phi_\xi\|_{\mathrm{Lip}}^2 \leq \beta_2 \int [\Phi_\xi - \Phi_\mu] \, d(\xi - \mu) \leq \beta_2 \|\Phi_\mu - \Phi_\xi\|_{\mathrm{Lip}} \|\xi - \mu\|_{\mathrm{KR}},
$$

and hence we have the desired result:

$$
\|\Phi_\mu - \Phi_\xi\|_{\mathrm{Lip}} \leq \beta_2 \|\xi - \mu\|_{\mathrm{KR}}.
$$

$\square$

**Lemma 7.** *Let* $K(x, y) = \exp(-\frac{\|x - y\|^2}{2\sigma^2})$. *Then* $\mathrm{MMD}_K^2(\mu, \nu) \leq \frac{1}{\sigma^2} \|\mu - \nu\|_{\mathrm{KR}}^2$.

*Proof.* First, we note that

$$
\begin{aligned}
\mathrm{MMD}_K^2(\mu, \nu) &= \int K(x, y) \, (\mu - \nu)(dx) \, (\mu - \nu)(dy) \\
&\leq \left[ \sup_{x \neq x'} \frac{\int (K(x, y) - K(x', y)) \, (\mu - \nu)(dy)}{d(x, x')} \right] \|\mu - \nu\|_{\mathrm{KR}} \\
&\leq \left[ \sup_{x \neq x', y \neq y'} \frac{K(x, y) - K(x, y') - K(x', y) + K(x', y')}{d(x, x') \, d(y, y')} \right] \|\mu - \nu\|_{\mathrm{KR}}^2.
\end{aligned}
$$

Next, we show that

$$c = \sup_{x \neq x', y \neq y'} \frac{K(x,y) - K(x,y') - K(x',y) + K(x',y')}{d(x,x')\, d(y,y')} = \frac{1}{\sigma^2}.$$

Because $K$ is differentiable, it suffices to check that the operator norm of $\partial_{x_i}\partial_{y_j} K(x,y)$ is bounded. To see this, note that

$$
\begin{aligned}
c &= \sup_{x \neq x'} \frac{\mathrm{Lip}\big(K(x,\cdot) - K(x',\cdot)\big)}{||x - x'||} \\
&= \sup_{x \neq x', y} \frac{||\nabla_y K(x,y) - \nabla_y K(x',y)||_2}{||x - x'||} \\
&\leq \sup_{x,y} ||\nabla_x \nabla_y K(x,y)||_{\mathrm{op}},
\end{aligned}
$$

where we obtained by the last equality by applying the vector-valued mean value theorem (Rudin, 1964) to $t \mapsto \nabla_y K((1-t)x + tx', y)$, thereby obtaining

$$
\begin{aligned}
||\nabla_y K(x,y) - \nabla_y K(x',y)||_2 &\leq ||\partial_t \nabla_y K((1-t)x + tx', y)||_2 \\
&= ||(x' - x)\cdot \nabla_x \nabla_y K((1-t)x + tx', y)||_2 \\
&\leq ||x' - x||_2 ||\nabla_x \nabla_y K((1-t)x + tx', y)||_{\mathrm{op}}.
\end{aligned}
$$

Now

$$
\begin{aligned}
\partial_{x_i}\partial_{y_j} K(x,y) &= \partial_{x_i}\partial_{y_j} \exp\left(-\frac{1}{2\sigma^2}\sum_k (x_k - y_k)^2\right) \\
&= \partial_{x_i} \exp\left(-\frac{1}{2\sigma^2}\sum_k (x_k - y_k)^2\right)\cdot -\frac{1}{\sigma^2}(x_j - y_j) \\
&= \exp\left(-\frac{1}{2\sigma^2}\sum_k (x_k - y_k)^2\right)\left[\frac{1}{\sigma^4}(x_i - y_i)(x_j - y_j) - \frac{1}{\sigma^2}\delta_{ij}\right].
\end{aligned}
\tag{30}
$$

This is the symmetric matrix

$$\frac{1}{\sigma^2}\exp\left(-\frac{||x-y||^2}{2\sigma^2}\right)\left[\frac{1}{\sigma^2}(x-y)(x-y)^T - I\right].$$

By inspection, we see that $x - y$ is an eigenvector with eigenvalue $\frac{1}{\sigma^2}\exp(-\frac{||x-y||^2}{2\sigma^2})(\frac{1}{\sigma^2}||x-y||^2 - 1)$, and the other eigenvectors are orthogonal to $x - y$ with eigenvalues $-\frac{1}{\sigma^2}\exp(-\frac{||x-y||^2}{2\sigma^2})$.

Setting $z = \frac{||x-y||^2}{2\sigma^2}$ and taking the absolute value of the eigenvalues, we see the maximum operator norm over all $x, y$ is equal to

$$\max\left\{\sup_{z \geq 0}\frac{1}{\sigma^2}e^{-z}|2z - 1|,\ \sup_{z \geq 0}\frac{1}{\sigma^2}e^{-z}\right\} = \frac{1}{\sigma^2}.$$

Therefore

$$c \leq \frac{1}{\sigma^2}.$$

It is not strictly necessary for the proof to show equality. However, to show equality, let $u$ be a unit vector, and set $x' = y' = 0$, and $x = y = tu$. Then

$$
\begin{aligned}
c &\geq \sup_{t \neq 0} \frac{K(tu, tu) - K(tu, 0) - K(0, tu) + K(0,0)}{d(tu, 0)\, d(tu, 0)} \\
&= \sup_{t \neq 0} \frac{2 - 2\exp(-\frac{t^2}{2\sigma^2})}{t^2} \\
&\geq \lim_{t \to 0} \frac{2 - 2\exp(-\frac{t^2}{2\sigma^2})}{t^2} \\
&= \lim_{t \to 0} \frac{-2\exp(-\frac{t^2}{2\sigma^2})\cdot -\frac{2t}{2\sigma^2}}{2t} \\
&= \frac{1}{\sigma^2},
\end{aligned}
$$

where we used l'Hôpital's rule to evaluate the limit. □

**Proposition 11.** *The minimax and non-saturating GAN losses do not satisfy (D3) for some $\mu_0$.*

*Proof.* For the minimax loss, the Bregman divergence is:

$$
\mathfrak{D}_{D_{\mathrm{JS}}(\cdot \,||\, \mu_0)}(\nu, \mu) = \int \left[ \frac{1}{2} \log \frac{\mu_0}{\frac{1}{2}\mu_0 + \frac{1}{2}\nu} \, d\mu_0 + \frac{1}{2} \log \frac{\nu}{\frac{1}{2}\mu_0 + \frac{1}{2}\nu} \, d\nu \right]
$$
$$
- \int \left[ \frac{1}{2} \log \frac{\mu_0}{\frac{1}{2}\mu_0 + \frac{1}{2}\mu} \, d\mu_0 + \frac{1}{2} \log \frac{\mu}{\frac{1}{2}\mu_0 + \frac{1}{2}\mu} \, d\mu \right]
$$
$$
- \int \frac{1}{2} \log \frac{\mu}{\mu_0 + \mu} \, d(\nu - \mu)
$$
$$
= \frac{1}{2} \int \log \frac{\mu_0 + \mu}{\mu_0 + \nu} \, d\mu_0 + \frac{1}{2} \int \left[ \log \frac{\nu}{\frac{1}{2}\mu_0 + \frac{1}{2}\nu} - \log \frac{\mu}{\mu_0 + \mu} \right] d\nu + \frac{1}{2} \log \frac{1}{2}
$$
$$
= \frac{1}{2} \int \log \frac{\mu_0 + \mu}{\mu_0 + \nu} \, d(\mu_0 + \nu) + \frac{1}{2} \int \log \frac{\nu}{\mu} \, d\nu
$$
$$
= D_{\mathrm{KL}}(\tfrac{1}{2}\nu + \tfrac{1}{2}\mu_0 \,||\, \tfrac{1}{2}\mu + \tfrac{1}{2}\mu_0) + \frac{1}{2} D_{\mathrm{KL}}(\nu \,||\, \mu),
$$

and for the non-saturating loss, the Bregman divergence is

$$
\mathfrak{D}_{D_{\mathrm{KL}}(\frac{1}{2}\cdot + \frac{1}{2}\mu_0 \,||\, \mu_0)}(\nu, \mu) = \int \log \frac{\frac{1}{2}\mu_0 + \frac{1}{2}\nu}{\mu_0} \, d(\tfrac{1}{2}\nu + \tfrac{1}{2}\mu_0)
$$
$$
- \int \log \frac{\frac{1}{2}\mu_0 + \frac{1}{2}\mu}{\mu_0} \, d(\tfrac{1}{2}\mu + \tfrac{1}{2}\mu_0)
$$
$$
+ \int \frac{1}{2} \log \frac{\mu_0}{\mu_0 + \mu} \, d(\nu - \mu)
$$
$$
= \frac{1}{2} \int \log \frac{\mu_0 + \nu}{\mu_0 + \mu} \, d\mu_0 + \frac{1}{2} \int \log \frac{\frac{1}{2}\mu_0 + \frac{1}{2}\nu}{\mu_0 + \mu} \, d\nu - \frac{1}{2} \log \frac{1}{2}
$$
$$
= D_{\mathrm{KL}}(\tfrac{1}{2}\nu + \tfrac{1}{2}\mu_0 \,||\, \tfrac{1}{2}\mu + \tfrac{1}{2}\mu_0).
$$

Choosing $\nu$ to be not absolutely continuous w.r.t. $\frac{1}{2}\mu + \frac{1}{2}\mu_0$ makes the Bregman divergence $\infty$, which is sufficient to show that the Bregman divergence is not bounded by $||\mu - \nu||_{\mathrm{KR}}^2$. □

**Proposition 12.** *The MMD loss with Gaussian kernel satisfies (D3) with $\beta_2 = 2\pi$ for all $\mu_0$.*

*Proof.* We work with the Gaussian kernel $K(x, y) = e^{-\frac{||x-y||^2}{2\sigma^2}}$ and use Proposition 10:

$$
\mathfrak{D}_{\frac{1}{2}\mathrm{MMD}^2(\cdot, \mu_0)}(\nu, \mu) = \frac{1}{2} \int K(x, y) \, (\nu - \mu_0)(dx) \, (\nu - \mu_0)(dy)
$$
$$
- \frac{1}{2} \int K(x, y) \, (\mu - \mu_0)(dx) \, (\mu - \mu_0)(dy)
$$
$$
- \int \big( K(x, y) \, \mu(dy) - K(x, y) \, \mu_0(dy) \big) \, (\nu - \mu)(dx)
$$
$$
= \frac{1}{2} \int K(x, y) \, (\nu - \mu)(dx) \, (\nu - \mu)(dy)
$$
$$
= \frac{1}{2} \mathrm{MMD}^2(\mu, \nu)
$$
$$
\leq \frac{1}{2\sigma^2} ||\mu - \nu||_{\mathrm{KR}}^2,
$$

where the last line is from Lemma 7. The result follows with $\sigma = (2\pi)^{-1/2}$.

The result actually applies more generally for the MMD loss with a differentiable kernel $K$ that satisfies $\beta_2 := \sup_{x,y} ||\nabla_x \nabla_y K(x, y)||_2 < \infty$. □

**Proposition 13** (Moreau–Yosida regularization). *Suppose $J : \mathcal{M}(X) \to \bar{\mathbb{R}}$ is convex, and define $\tilde{J} := J \oplus R_3$. Then $\tilde{J}$ is convex, and $\mathfrak{D}_{\tilde{J}}(\nu, \mu) \leq \frac{\beta_2}{2}||\mu - \nu||^2_{\mathrm{KR}}$.*

*Proof.* We work on the more general case of $K(x, y) = (2\pi\sigma^2)^{-d/2} \exp(-\frac{||x-y||^2}{2\sigma^2})$, and define

$$\tilde{J}(\mu) = \inf_{\tilde{\mu}} J(\tilde{\mu}) + \frac{cL}{2}\mathrm{MMD}^2_K(\mu, \tilde{\mu})$$

for $c = \sigma^2(2\pi\sigma^2)^{d/2}$.

Let $\mu^*$ be the unique minimizer of the infimum, which exists because the function is a strongly convex. By the envelope theorem, we compute that

$$
\begin{aligned}
\frac{d}{d\epsilon}\tilde{J}(\mu + \epsilon\chi)\Big|_{\epsilon=0} &= \frac{d}{d\epsilon}\frac{cL}{2}||\mu + \epsilon\chi - \mu^*||^2_{\mathcal{H}}\Big|_{\epsilon=0} \\
&= \frac{d}{d\epsilon}\frac{cL}{2}\left(||\mu - \mu^*||^2_{\mathcal{H}} + 2\epsilon\langle\mu - \mu^*, \chi\rangle_{\mathcal{H}} + \epsilon^2||\chi||^2_{\mathcal{H}}\right)\Big|_{\epsilon=0} \\
&= cL\langle\mu - \mu^*, \chi\rangle_{\mathcal{H}} \\
&= cL\int(\mu - \mu^*)\,d\chi,
\end{aligned}
$$

so $\frac{\delta\tilde{J}}{\delta\mu} = cL(\mu - \mu^*)$.

Then

$$
\begin{aligned}
\mathfrak{D}_{\tilde{J}}(\nu, \mu) &= \tilde{J}(\nu) - \tilde{J}(\mu) - \int\frac{\delta\tilde{J}}{\delta\mu}\,d(\nu - \mu) \\
&= \inf_{\tilde{\nu}} J(\tilde{\nu}) + \frac{cL}{2}||\nu - \tilde{\nu}||^2_{\mathcal{H}} - \left[J(\mu^*) + \frac{cL}{2}||\mu - \mu^*||^2_{\mathcal{H}}\right] - cL\langle\mu - \mu^*, \nu - \mu\rangle_{\mathcal{H}} \\
&\leq J(\mu^*) + \frac{cL}{2}||\nu - \mu^*||^2_{\mathcal{H}} - \left[J(\mu^*) + \frac{cL}{2}||\mu - \mu^*||^2_{\mathcal{H}}\right] - cL\langle\mu - \mu^*, \nu - \mu\rangle_{\mathcal{H}} \\
&= \frac{cL}{2}||\mu - \nu||^2_{\mathcal{H}} \\
&\leq \frac{cL}{2}\cdot(2\pi\sigma^2)^{-d/2}\cdot\frac{1}{\sigma^2}||\mu - \nu||^2_{\mathrm{KR}} \\
&= \frac{L}{2}||\mu - \nu||^2_{\mathrm{KR}},
\end{aligned}
$$

where we used Lemma 7 for the second-to-last line. The proposition follows for $\sigma = (2\pi)^{-1/2}$.

Regarding this choice of $\sigma$, it will turn out that in the general case, the dual penalty includes a numerically unfavorable factor of $(2\pi\sigma^2)^{-d/2}$, dependent on the dimension of the problem. In practical applications, such as image generation, $d$ can be quite large, making the accurate computation of $(2\pi\sigma^2)^{-d/2}$ completely infeasible. For numerical stability, we propose choosing the critical parameter $\sigma = (2\pi)^{-1/2}$, which corresponds to the dimension-free kernel $K(x, y) = e^{-\pi||x-y||^2}$. $\qquad\square$

**Lemma 8.** *Let $\mathcal{H}$ be an RKHS with a continuous kernel $K$ on a compact domain $X$. The convex conjugate of $\mu \mapsto \frac{\lambda}{2}||\mu||^2_{\mathcal{H}}$ is $f \mapsto \frac{1}{2\lambda}||f||^2_{\mathcal{H}} + \chi\{f \in \mathcal{H}\}$.*

*Proof.* First we show an upper bound for $f \in \mathcal{H}$:

$$
\sup_{\mu} \left[ \int f \, d\mu - \frac{\lambda}{2} \|\mu\|_{\mathcal{H}}^2 \right] = \sup_{\mu} \left[ \langle f, \mu \rangle_{\mathcal{H}} - \frac{\lambda}{2} \|\mu\|_{\mathcal{H}}^2 \right]
$$

$$
\leq \sup_{\mu} \left[ \|f\|_{\mathcal{H}} \|\mu\|_{\mathcal{H}} - \frac{\lambda}{2} \|\mu\|_{\mathcal{H}}^2 \right]
$$

$$
\leq \sup_{z \in \mathbb{R}} \left[ \|f\|_{\mathcal{H}} z - \frac{\lambda}{2} z^2 \right]
$$

$$
= \|f\|_{\mathcal{H}} \cdot \frac{\|f\|_{\mathcal{H}}}{\lambda} - \frac{\lambda}{2} \cdot \frac{\|f\|_{\mathcal{H}}^2}{\lambda^2}
$$

$$
= \frac{1}{2\lambda} \|f\|_{\mathcal{H}}^2.
$$

To derive a lower bound for general $f \in \mathcal{C}(X)$, we use the Mercer decomposition of the positive definite kernel $K$,

$$
K(x, y) = \sum_{i=1}^{\infty} \gamma_i \phi_i(x) \phi_i(y)
$$

where $\gamma_i \geq 0$ are eigenvalues and $\{\phi_i\}_{i=1}^{\infty}$ is a complete orthonormal sequence of $L^2(X; dx)$. It is well-known that the corresponding RKHS is given by

$$
\mathcal{H} = \left\{ g \in L^2(X, dx) \mid \sum_{i=1}^{\infty} \frac{(g, \phi_i)_{L^2(X)}^2}{\gamma_i} < \infty \right\},
$$

and the norm of $g \in \mathcal{H}$ by

$$
\|g\|_{\mathcal{H}}^2 = \sum_{j=1}^{\infty} \frac{(g, \phi_i)_{L^2(X)}^2}{\gamma_i}.
$$

Let $f \in \mathcal{C}(X) \subset L^2(X, dx)$ be an arbitrary function. We replace the supremum of the conjugate function

$$
\sup_{\mu} \left[ \int f \, d\mu - \frac{\lambda}{2} \|\hat{\mu}\|_{\mathcal{H}}^2 \right] \tag{31}
$$

by $\mu \in \mathcal{M}(X)$ that has a square-integrable Radon–Nykodim derivative with respect to the Lebesgue measure $dx$, and then we have a lower bound. We use $\mu(x)$ for the Radon–Nykodim derivative with slight abuse of notation.

Suppose $f(x) = \sum_{j=1}^{\infty} a_j \phi_j(x)$ and $\mu(x) = \sum_{j=1}^{\infty} b_j \phi_j(x)$ are the expansion. Note that, since the kernel embedding is given by

$$
\hat{\mu}(x) = \int K(x, y) \mu(y) \, dy = \sum_{j=1}^{\infty} \gamma_j b_j \phi_j(x),
$$

the above maximization is reduced to

$$
\sup_{b_j} \sum_{j=1}^{\infty} a_j b_j - \frac{\lambda}{2} \gamma_j b_j^2.
$$

This is maximized when $b_j = a_j / (\lambda \gamma_j)$, and the maximum value is

$$
\frac{1}{2\lambda} \sum_{j=1}^{\infty} \frac{a_j^2}{\gamma_j}. \tag{32}
$$

If $f \in \mathcal{H}$, this value is finite, and the lower bound of the conjugate given by is $\frac{1}{2\lambda} \|f\|_{\mathcal{H}}^2$, which is the same as the upper bound. If $f \notin \mathcal{H}$, the value (32) is infinite, and the conjugate function takes $+\infty$. $\qquad \square$

**Proposition 14.** *Let $\mathcal{H}$ be an RKHS with the Gaussian kernel $K(x,y) = e^{-\pi||x-y||^2}$. Then for $f \in \mathcal{H}$,*

$$||f||_{\mathcal{H}}^2 = \sum_{k=0}^{\infty} (4\pi)^{-k} \sum_{k_1+\cdots+k_d=k} \frac{1}{\prod_{i=1}^{d} k_i!} ||\partial_{x_1}^{k_1} \cdots \partial_{x_d}^{k_d} f||_{L^2(\mathbb{R}^d)}^2 \tag{21}$$

$$= ||f||_{L^2(\mathbb{R}^d)}^2 + \frac{1}{4\pi}||\nabla f||_{L^2(\mathbb{R}^d)}^2 + \frac{1}{16\pi^2}||\nabla^2 f||_{L^2(\mathbb{R}^d)}^2 + \textit{other terms}. \tag{22}$$

*Proof.* We consider the more general case of $K(x,y) = (2\pi\sigma^2)^{-d/2} e^{-||x-y||^2/2\sigma^2}$, where we have

$$||f||_{\mathcal{H}}^2 = \sum_{k=0}^{\infty} (\tfrac{1}{2}\sigma^2)^k \sum_{k_1+\cdots+k_d=k} \frac{1}{\prod_{i=1}^{d} k_i!} ||\partial_{x_1}^{k_1} \cdots \partial_{x_d}^{k_d} f||_{L^2(\mathbb{R}^d)}^2$$

$$= ||f||_{L^2(\mathbb{R}^d)}^2 + \tfrac{1}{2}\sigma^2||\nabla f||_{L^2(\mathbb{R}^d)}^2 + \tfrac{1}{4}\sigma^4||\nabla^2 f||_{L^2(\mathbb{R}^d)}^2 + \text{other terms}.$$

Novak et al. (2018) prove this result for $\sigma = 1$. We sketch the proof for the general case here. We use the Fourier transform convention that

$$\hat{f}(k) = (2\pi)^{-d/2} \int_{\mathbb{R}^d} f(x) e^{-ik \cdot x} \, dx, \quad f(x) = (2\pi)^{-d/2} \int_{\mathbb{R}^d} \hat{f}(k) e^{ik \cdot x} \, dk.$$

Consider the inner product

$$\langle f, g \rangle = \int_{\mathbb{R}^d} e^{\sigma^2 ||k||^2/2} \, \hat{f}(k) \, \overline{\hat{g}(k)} \, dk,$$

defined for functions $||f|| < \infty$. Expanding the exponential in Taylor series, this inner product gives the equation for the norm in the proposition. By use of the Fourier inversion formula, it can be shown that $\langle f, K_x \rangle = f(x)$, where

$$K_x(y) = (2\pi\sigma^2)^{-d/2} e^{-||x-y||^2/2\sigma^2},$$

so this is an RKHS with the Gaussian kernel. $\qquad\square$

