# OpenReview forum: "Smoothness and Stability in GANs"
_ICLR.cc/2020/Conference — Accept (Poster)_

### Official Review · AnonReviewer1 · 2019-10-15
**Official Blind Review #1**

**Rating:** 1

**Review:**

The paper provides new theoretical view on GAN regularisation. However, it lacks proper empirical evaluation and makes an impression of a work in progress. Furthermore, the conclusions lead mostly to common techniques that have already been studied.

Pros:
- Theorem 1 provides sufficient conditions for convergence of generator gradients to zero (under assumption of optimal discriminators).
- New view on combining loss functions and regularizers via inf-convolutions.
- Clarification of a difference between gradient penalties and spectal normalization.

Cons:
- No evaluation with respect to any reasonable GAN setting.
- Proposed regularization technique combines existing methods and does not actually propose new ones.
- The main insights of sections 4 and 5 are trivial, like enforcing Lipshitzness of optimal discriminator by optimization of only Lipshitz discriminators.
- It is unclear wheather proposed solutions are practical, e.g. use of smooth activation functions may be costly and may lead to vanishing gradients. Again, experiments would be desired.
- Same combination of regularization techniques (gradient penalty, spectral norm and MMD loss) has been studied by [1] in various forms (Gradient-Constrained MMD, Scaled MMD). However, there is no discussion of similarities and differences between these works.
- Submission's main text is 10 pages long without sufficient reasons for that (figures, tables).

Detailed comments:
(1) End of Section 4: 'Theorem 1 also suggests that applying only Lipschitz constraints is not enough to stabilize GANs'. Theorem 1 is not 'iff', so Lipshitz constraint *may be* not enough.
(2) Section 6 concludes that penalization of discriminators RKHS's norm is required. It is unclear, however, why discriminator function would belong to such space.
(3) In Appendix B authors say, in the context of WGAN, that 'The Lipschitz constraint on the discriminator is typically enforced by spectral normalization (Miyato et al., 2018), (...)'. This setting fails, as stated earlier in the Introduction.
(4) It seems there is conceptual misundersting of what MMD-GANs are in Appendix B. Authors say 'Despite their names, MMD-GANs (Li et al., 2017a; Arbel et al., 2018) typically do not directly minimize the MMD but instead an adversarial version of the MMD'. GANs by definition are adversarial, while optimization against MMD alone is not. Hence, it is *according to their names*, not 'despite'.
Generator losses implied by MMD-GANs under assumption of optimal discriminators, have been termed 'Optimized MMD' [1] and studied earlier in [2].
(5) Given (4), The Table 2. includes MMD as a GAN loss, although authors probably refer to the properties of non-adversarial Generative Moment Matching Networks [3].


[1] Michael Arbel, Dougal Sutherland, Mikołaj Binkowski, and Arthur Gretton. On gradient regularizers for MMD GANs. In Advances in Neural Information Processing Systems, pp. 6700–6710, 2018.
[2] B. K. Sriperumbudur, K. Fukumizu, A. Gretton, G. R. G. Lanckriet, and B. Schölkopf. “Kernel choice and classifiability for RKHS embeddings of probability distributions.” NIPS. 2009
[3] Yujia Li, Kevin Swersky, Richard Zemel, "Generative Moment Matching Networks", ICML 2015.

**Experience Assessment:**

I have published one or two papers in this area.

**Review Assessment: Checking Correctness Of Derivations And Theory:**

I assessed the sensibility of the derivations and theory.

**Review Assessment: Checking Correctness Of Experiments:**

I carefully checked the experiments.

**Review Assessment: Thoroughness In Paper Reading:**

I read the paper thoroughly.

---

> ### Author Response · Authors · 2019-11-11
> **Response to Official Review #1**
>
> Thank you for your time in writing in-depth feedback. We have read your comments carefully and have found them very helpful in guiding our revisions. Please find our response to your primary concerns below, as well as our more detailed response afterwards (due to the character limit):
>
>
> > No evaluation with respect to any reasonable GAN setting.
> > Proposed regularization technique combines existing methods and does not actually propose new ones.
>
> We would like to clarify the aim of our paper, since there appears to be a mismatch between our intended message and the reviewer’s vision for our project. Most importantly, we are not trying to propose a new GAN variant or new regularization techniques. Instead, we explain the need for and the sensibility of existing GAN techniques from the unifying framework of a desire for smoothness, thereby placing the use of these techniques on firm theoretical footing. This is a novel perspective not expressed by previous work to our knowledge. We kindly ask that our paper be evaluated with this aim in mind, not from the perspective of proposing and testing a new GAN variant.
>
>
> > The main insights of sections 4 and 5 are trivial, like enforcing Lipshitzness of optimal discriminator by optimization of only Lipshitz discriminators.
>
> We agree that the strategy for practically enforcing that the optimal discriminator be Lipschitz (section 4) and smooth (section 5) is obvious, namely, to only optimize over the relevant set of discriminators.
>
> However, the main insight we are intending to share in these sections is not *how* to practically constrain the optimal discriminator, but *why* constraining the optimal discriminator is a sensible thing to do in the first place. Our analysis finds that it is indeed sensible because the process preserves the minimizers of the original loss function $J$. This is because constraining the optimal discriminator is equivalent to inf-convolving the original loss function $J$ with a regularizer $R$, which preserves the set of minimizers. This argument requires carefully reasoning through the interplay between the regularizer $R$ (which determines the loss function) and its convex conjugate $R^\star$ (which determines the properties of the optimal discriminator). We acknowledge that this point may have been lost in the formalism, and we are updating the draft to clarify this point.
>
>
> > It is unclear wheather proposed solutions are practical, e.g. use of smooth activation functions may be costly and may lead to vanishing gradients. Again, experiments would be desired.
>
> We agree with you and Reviewer 3 that a study of smooth activation functions would be insightful. We are considering what steps to take to address this issue.

---

> > ### Author Response · Authors · 2019-11-11
> > **Response to Official Review #1 (continued)**
> >
> > > Same combination of regularization techniques (gradient penalty, spectral norm and MMD loss) has been studied by [1] in various forms (Gradient-Constrained MMD, Scaled MMD). However, there is no discussion of similarities and differences between these works.
> >
> > Arbel et al. obtain strong theoretical and empirical results using a combination of techniques that features many of the same ingredients as we derive in our analysis. Interestingly, these techniques serve different purposes in their work compared to ours. In their work, spectral normalization is used to improve the conditioning of the critic rather than to constrain its Lipschitz constant, as in our analysis. Regarding gradient penalties and MMD, in their work, gradient norms are combined with MMD (with a learned kernel) to obtain a novel discrepancy measure, whereas we show that regularizing an arbitrary loss with a Gaussian-kernel MMD leads to gradient penalties.
> >
> >
> > > (1) End of Section 4: 'Theorem 1 also suggests that applying only Lipschitz constraints is not enough to stabilize GANs'. Theorem 1 is not 'iff', so Lipshitz constraint *may be* not enough.
> >
> > We tried to be careful about the wording (‘suggests’ rather than ‘implies’), but we will rephrase this to make it clear. We are editing the end of Section 2 to emphasize this point, that it is possible that our analysis is too conservative.
> >
> >
> > > (2) Section 6 concludes that penalization of discriminators RKHS's norm is required. It is unclear, however, why discriminator function would belong to such space.
> >
> > In Section 6, we have adopted the convention that if $f$ is not in $\mathcal{H}$, then $|| f ||_{\mathcal{H}}$ is infinite. This is a point we will clarify, but it is made rigorous by Lemma 6. Although the optimal discriminator of the *original* loss function may not belong to the RKHS, the optimal discriminator of the loss function regularized by $R_3$ is guaranteed to belong to the RKHS (due to Lemma 6 and equation 4 [equation 20 in the updated draft]).
> >
> >
> > > (4) It seems there is conceptual misundersting of what MMD-GANs are in Appendix B. Authors say 'Despite their names, MMD-GANs (Li et al., 2017a; Arbel et al., 2018) typically do not directly minimize the MMD but instead an adversarial version of the MMD'. GANs by definition are adversarial, while optimization against MMD alone is not. Hence, it is *according to their names*, not 'despite'. Generator losses implied by MMD-GANs under assumption of optimal discriminators, have been termed 'Optimized MMD' [1] and studied earlier in [2].
> > > (5) Given (4), The Table 2. includes MMD as a GAN loss, although authors probably refer to the properties of non-adversarial Generative Moment Matching Networks [3].
> >
> > Thank you for bringing these points to our attention. We now realize this is confusing phrasing and will modify the text accordingly. It seems that the crux of the disagreement is that in this paper, we restrict our attention to the minimization of any convex function of a probability distribution, which is always equivalent to some adversarial game according to equation (3) [equation (4) in the updated draft]. When we consider MMD as a loss, we are indeed referring to GMMN; under our framework, GMMN is adversarial, with the optimal discriminator approximated by samples rather than by training a separate neural network. This is why we listed it in Table 1 as a “GAN.” We had also listed MMD-GAN alongside GMMN in Table 1 because they coincide in the special case of a single kernel, for the benefit of readers who might be unfamiliar with GMMN but have heard of MMD-GAN, but we now acknowledge this may be misleading. Regarding the “despite their names” comment, we are referring to the “MMD” part, not the “GAN” part: it might be expected that MMD-GANs minimize the MMD instead of the Optimized MMD, in analogy with Wasserstein GANs, which minimize the Wasserstein distance, and f-GANs, which minimize an f-divergence.

---

### Official Review · AnonReviewer2 · 2019-10-20
**Official Blind Review #2**

**Rating:** 6

**Review:**

The work studies the relationship between the stability and the smoothness of GANs based on the proposition which was proposed by Bertsekas . It explains many nontrivial empirical observations when one is training GANs, including both of the necessities of the spectral normalization and the gradient penalty, in a theoretical perspective. And the work points out that most common GAN losses do not satisfy the all of the smoothness conditions, thereby corroborating their empirical instability. Meanwhile, it develops regularization techniques that enforce the smoothness conditions, which can lead to stability of the GAN.

Pros
1. The paper theoretically gives a reasonable explanation of why applying a gradient penalty together spectral norm seems to improve performance of generator.
2.  The proofs of the theorems and the propositions in this paper are gorgeous and beautiful.

Cons
1. As the paper concludes, in practice, it is impossible to let the generator be trained after the discriminator attain theoretical optimal. As a paper which topic is about the training process of GAN, it is better to account for real situation.

2. The experiment section is too simple and lacks of persuasiveness. The main theorem only gives the sufficiency of those conditions. I think it’s necessary to give an example which can imply that anyone condition is essential.

3. Proposition 9, Proposition 12 and Equation (7) show the equivalence between the condition (D3) and the existence of the regularization term of the reproducing kernel Hilbert space norm of the discriminator. But after this, the paper uses the first order term of the expansion in Proposition 13 to substitute $\|\psi\|_{H}^2$. The condition (D3) doesn't necessarily still hold if only adding the gradient penalty term to the objective function. Why it can be supposed that the first order term of the expansion plays a leading role in penalizing ? Isn't it unconvincing to explain the necessity of the gradient penalty from the perspective of making the condition (D3) true?

**Experience Assessment:**

I have published one or two papers in this area.

**Review Assessment: Checking Correctness Of Derivations And Theory:**

I assessed the sensibility of the derivations and theory.

**Review Assessment: Checking Correctness Of Experiments:**

I assessed the sensibility of the experiments.

**Review Assessment: Thoroughness In Paper Reading:**

I read the paper at least twice and used my best judgement in assessing the paper.

---

> ### Author Response · Authors · 2019-11-13
> **Response to Review #2**
>
> Thank you for your feedback and kind words regarding our proofs! We are pleased that the proofs neatly tie together concepts from convex analysis, optimal transport, and RKHS theory, and we hope they inspire future proof techniques in this area.
>
>
> > 1. As the paper concludes, in practice, it is impossible to let the generator be trained after the discriminator attain theoretical optimal. As a paper which topic is about the training process of GAN, it is better to account for real situation.
>
> This is unfortunately a disadvantage of the approach taken by this and prior works (please see the related work section). We hope that future work in our field will further bridge this gap between theory and practice.
>
>
> > 2. The experiment section is too simple and lacks of persuasiveness.
>
> Because the main contribution of this submission is a rigorous theoretical framework on the stability of GAN training, we wanted to choose a setting where we could numerically evaluate our theory, which required choosing a generator where we could analytically compute the relevant Lipschitz constants. The experimental result, while simple, supports the theoretical implication.
>
>
> > The condition (D3) doesn't necessarily still hold if only adding the gradient penalty term to the objective function. Why it can be supposed that the first order term of the expansion plays a leading role in penalizing ? Isn't it unconvincing to explain the necessity of the gradient penalty from the perspective of making the condition (D3) true?
>
> Recall that each penalty term in the infinite series encourages an additional degree of regularity on the optimal discriminator, and the regularity of the optimal discriminator corresponds to the regularity of the implied loss function $J$ being minimized, via duality. It is correct that without all the terms of the infinite series, D3 is not guaranteed to be satisfied. When fewer penalty terms are used, the regularization effect on the implied loss function is reduced, but the penalty terms that are present will still encourage partial regularity of the implied loss function. We will add a remark on this matter to the end of Section 6. We view the choice of only using the leading terms as a disadvantageous but practical necessity.

---

### Official Review · AnonReviewer3 · 2019-10-22
**Official Blind Review #3**

**Rating:** 8

**Review:**

This paper provides a unified theoretical framework for regularizing GAN losses. It accounts for most regularization technics especially spectral normalization and gradient penalty and explains how those two methods are in fact complementary. So far this was only observed experimentally but without any theoretical insight. The result goes beyond that as the criterion could be applied to general convex cost functional.
The main general theorem is Theorem 1 which states 3 conditions on the optimal critic and 2 others on the generator. The paper is mainly concerned by the conditions on the optimal critic and show that the first 2 conditions can be achieved by the Spectral normalization, while the last one can be achieved by some gradient penalty.
The paper is clearly written, well structured and pleasant to read.
I have the following two remarks:
	- Proposition 8 provides a way to ensure condition 2 holds (beta-smoothness). It requires spectral normalization and smooth activation functions. In practice, while the spectral normalization is important, the choice of the activation is not in general 1-smooth (Leaky-relu for instance). Does it really matter in practice?
	Some illustrative experiments could be beneficial to better understand what's happening.
	- Is it that hard to obtain generators that satisfy condition G1 and G2, it seems to be a natural consequence on the regularity of the mapping f? If that is the case, it might be worth better explaining how this is challenging.

Limitations: The paper considers only the setting where the optimal critic is reached and therefore it is still unclear if the analysis carries on to the training procedures used in practice (non-optimal critic). The authors recognize this limitation and leave it for future work.

Overall, I feel that the paper provides good insights on what regularization is important for training gans and why. For that reason, I think this paper should be accepted.


------------------------------------------------------------------------------------------------------------
Revision:

I think the paper provides a good theoretical contribution in terms of interpreting many of the tricks used for improving GAN training. In fact the paper also suggests some new regularization methods (prop 13 for conditions D3)  which would constrain the RKHS norm of the critic. The authors show how it is related to gradient penalty, in a particular case, but the result also suggests something more general. For instance [1], consider an abstract RKHS space containing deep networks and provide an upper-bound on the rkhs norm of such networks in terms of the spectral norm of their weights and a lower-bound in terms of its Lipschitz constant.

I do agree with reviewer 1 that a better discussion of the connection to [2] should be included since that paper was interested in  ensuring weak continuity of the loss, which can be thought of as  a first requirement to get more regularity of the cost functional.

I still think the paper is worth being accepted and raised my score to 8 as I think the authors addressed the major concerns that were raised.

[1] A. Bietti, G. Mialon, D. Chen, and J. Mairal. A Kernel Perspective for Regularizing Deep Neural Networks.
[2] Michael Arbel, Dougal Sutherland, Mikołaj Binkowski, and Arthur Gretton. On gradient regularizers for MMD GANs. In Advances in Neural Information Processing Systems, pp. 6700–6710, 2018.







**Experience Assessment:**

I have published one or two papers in this area.

**Review Assessment: Checking Correctness Of Derivations And Theory:**

I assessed the sensibility of the derivations and theory.

**Review Assessment: Checking Correctness Of Experiments:**

I assessed the sensibility of the experiments.

**Review Assessment: Thoroughness In Paper Reading:**

I read the paper at least twice and used my best judgement in assessing the paper.

---

> ### Author Response · Authors · 2019-11-14
> **Response to Review #3**
>
> Thank you for your feedback. We are happy to hear that you found the paper insightful and pleasant to read. Please find responses to your questions below:
>
>
> > Proposition 8 provides a way to ensure condition 2 holds (beta-smoothness). It requires spectral normalization and smooth activation functions. In practice, while the spectral normalization is important, the choice of the activation is not in general 1-smooth (Leaky-relu for instance). Does it really matter in practice? Some illustrative experiments could be beneficial to better understand what's happening.
>
> We agree that it would be an insightful experiment to understand how stability is affected by a choice of non-smooth activation functions. In the case of ReLU or LeakyReLU, the discontinuity at 0 makes the function non-smooth, but we conjecture that Proposition 8 [Proposition 9 in the updated draft] and our stability results may still hold in some approximate sense, since these activations can be well-approximated by smooth functions (e.g. $\frac{1}{4}\log (1+e^{4x})$).
>
>
> > Is it that hard to obtain generators that satisfy condition G1 and G2, it seems to be a natural consequence on the regularity of the mapping f? If that is the case, it might be worth better explaining how this is challenging.
>
> Thank you for this suggestion. It is true that if the generator $f_\theta(z)$ has bounded first and second derivatives with respect to $\theta$, then it will satisfy conditions G1 and G2 for some constants $A$ and $B$. However, recall that these constants dictate how small the learning rate must be to guarantee stability, via Proposition 1. Thus, in order to obtain non-vacuous claims of stability with learning rates used in practice, it is not useful to simply claim that $A$ and $B$ are finite; instead, it is important to compute tight bounds for $A$ and $B$. These computations will vary quite a bit with the choice of architecture used (feedforward, convolutional, ResNet, etc.) and may lead to new generator architectures and regularization techniques. Due to the complexity of these computations and the neat logical separation of discriminator and generator allowed by Theorem 1, we think these computations are best suited for future work. We will add a remark explaining this matter at the end of Section 2.

---

### Author Response · Authors · 2019-11-15
**Response to all reviewers: Paper update**

We would like to again thank our reviewers for their valuable comments. We have updated our paper based on their feedback. The major updates are as follows:

- We would like to reiterate that our main purpose is to provide firmly rooted theoretical justification for commonly used GAN regularization techniques, by means of a novel theoretical framework based on smoothness and convex duality. We have polished Sections 2 and 3 to make sure the focus and argument are clear.

- A major advantage of the inf-convolution-based regularization framework is that it injects the desired regularity conditions without changing the minimizer of the original objective. We have highlighted a theoretically non-trivial result on minimizer invariance in Section 3 to emphasize this point. Please see our response to Review #1 for details.

---

### Decision · Program_Chairs · 2019-12-19

**Decision:**

Accept (Poster)

**Comment:**

The paper provides a theoretical study of what regularizations should be used in GAN training and why. The main focus is that the conditions on the discriminator that need to be enforced, to get the Lipshitz property of the corresponding function that is optimized for the generator. Quite a few theorems and propositions are provided. As noted by Reviewer3, this adds insight to well-known techniques: the Reviewer1 rightfully notes that this does not lead to any practical conclusion.
Moreover, then training of GANs never goes to the optimal discriminator, that could be a weak point; rather than it proceeds in the alternating fashion, and then evolution is governed by the spectra of the local Jacobian (which is briefly mentioned). This is mentioned in future work, but it is not clear at all if the results here can be helpful (or can be generalized).
 At some point of the paper it gets to "more theorems mode" which make it not so easy and motivating to read.
The theoretical results at the quantitative level are very interesting.  I have looked for a long time on Figure 1: does this support the claims? First my impression was it does not (there are better FID scores for larger learning rates). But in the end, I think it supports: the convergence for a smaller that $\gamma_0$ learning rate to the same FID indicated the convergence to the same local minima (probably). This is perfectly fine. Oscillations afterwards move us to a stochastic region, where FID oscillates. So, the theory has at least minor confirmation.